# Subwavelength dielectric waveguide for efficient travelling-wave magnetic resonance imaging

Yang Gao [1,2,3] ✉, Tong Liu[1], Tao Hong[1,2], Youtong Fang[3], Wen Jiang[1,2] & Xiaotong Zhang [3,4,5,6] ✉

Magnetic resonance imaging (MRI) has diverse applications in physics, biology, and medicine. Uniform excitation of nuclei spins through circular-polarized transverse magnetic component of electromagnetic field is vital for obtaining unbiased tissue contrasts. However, achieving this in the electrically large human body poses a significant challenge, especially at ultra-high fields (UHF) with increased working frequencies (≥297 MHz). Canonical volume resonators struggle to meet this challenge, while radiative excitation methods like travelling-wave (TW) show promise but often suffer from inadequate excitation efficiency. Here, we introduce a new technique using a sub-wavelength dielectric waveguide insert that enhances both efficiency and homogeneity at 7 T. Through TE11-to-TM11 mode conversion, power focusing, wave impedance matching, and phase velocity matching, we achieved a 114% improvement in TW efficiency and mitigated the center-brightening effect. This fundamental advancement in TW MRI through effective wave manipulation could promote the electromagnetic design of UHF MRI systems.

Uniform excitation of nuclei spins through circular-polarized transverse magnetic component of electromagnetic field $\mathbf{B}_1^+ = \mu \cdot (H_x + iH_y)/2$ is crucial for producing unbiased tissue contrasts in magnetic resonance imaging (MRI)[1–6], which is especially important for clinical diagnosis[7,8]. Because of higher $\mathbf{B}_1^+$ working frequency required at ultra-high field (UHF, 7 T and above) MRI systems, electrically large human body makes this goal difficult to achieve[9,10]. Thus, UHF MRI applications in clinical scenarios have been hindered for decades. According to the Lamor Frequency requirement for hydrogen spin resonance, the working frequency of $\mathbf{B}_1^+$ field at UHF is beyond ~297 MHz, located in the radio-frequency (RF) range. For the example of 7 T MRI, the human head size surpasses the corresponding wavelength in biological materials. Under the circumstance, the standing-wave (SW) excitation at reactive near regions, which is traditionally achieved with a birdcage resonator[11,12], cannot guarantee uniform distribution of $\mathbf{B}_1^+$ inside the human body due to the deviation from the quasi-static condition; conversely, it produces the well-known center-brightening effect[13] as well as dark spot, which respectively indicate the presence of SW node and antinode. Therefore, the birdcage resonators which have been conventionally equipped as whole-body transmit coils in clinical MRI systems (1.5 T and 3 T), may not produce uniform excitations over large field-of-view (FOV) at UHF.

Despite relentless efforts in resonator designs[14–17] and proactive $\mathbf{B}_1^+$ control techniques such as RF shimming via phased arrays[18,19], permittivity pads[20,21], and metamaterials[22,23], RF transmitters for UHF MRI are mostly designed or arranged in a volume shape where SW

[1]Hangzhou Institute of Technology, Xidian University, Hangzhou, China. [2]School of Electronic Engineering, National Key Laboratory of Antennas and Microwave Technology, Xidian University, Xi'an, China. [3]College of Electrical Engineering, Zhejiang University, Hangzhou, China. [4]Second Affiliated Hospital of Zhejiang University School of Medicine, Hangzhou, China. [5]MOE Frontier Science Center for Brain Science and Brain-machine Integration, Zhejiang University, Hangzhou, China. [6]Interdisciplinary Institute of Neuroscience and Technology, School of Medicine, Zhejiang University, Hangzhou, China. ✉e-mail: gaoyang01@xidian.edu.cn; zhangxiaotong@zju.edu.cn

always resides[24], and uniform excitation over an electrically large region is hence difficult to achieve. As a result, although with additional degrees of freedom, complex $B_1^+$ control methods with multi-channel transmission system can mostly excite uniform area in a limited FOV. Besides, such approach elevates the risk of specific absorption rate (SAR), increases system complexity[25], and imposes negative impact over MR signal acquisition stability under subject motion[26]. Due to above limitations, complex $B_1^+$ control systems are only allowed to operate under the research mode (multi-channel transmission). Because of limited access to the multi-channel transmission system in clinical scenarios, feasible RF excitation method which is able to produce uniform $B_1^+$ under the clinical mode (single-channel transmission) remains in urgent need. Recent advances in coupled-mode methods[27,28] as well as power splitters[29–32] have shown the feasibility of driving phased array resonators under the single-channel mode with improved $B_1^+$ homogeneity.

Compared to SW, travelling wave (TW) is naturally more uniform in magnitude when propagating through an infinitely large homogenous medium. For the simple format of a traveling wave $f_{TW}$, it can be explicitly expressed as the function of angular frequency ω, spatial coordinate $r$ and time $t$: $f_{TW}(r, t) = \sin(\omega \cdot t + |k| \cdot r)$, while a standing wave $f_{SW}$ can be expressed as the superposition of two TWs propagating in opposite directions: $f_{SW}(r,t) = f_{TW}^+ + f_{TW}^- = \sin(\omega \cdot t) \cdot \cos(|k| \cdot r)$. It is clear that the magnitude of TW is a constant, while the magnitude of SW varies in space as the function of $\cos(|k| \cdot r)$, where $|k| = 2\pi/\lambda$ is the wave number along propagation direction, $\lambda$ is the wavelength. $B_1^+$ inhomogeneity residing in SW scales up with increased working frequency.

The advantage of TW excitation in producing uniform $B_1^+$ in MRI has been discovered decades ago through modifying a canonical birdcage resonator into a TW antenna[33]. Recently, a leak-wave antenna, which is another version of TW antenna, has been reported[34]. The intrinsic low SAR characteristic of TW excitation has been disclosed. However, additional terminators as well as dielectric materials are required to produce TW current distribution for both approaches, and samples have to be placed in close vicinity of the antenna (at reactive near-field region) to ensure efficient power transmission, otherwise most power will be dissipated in terminators.

The first implementation of TW excitation at radiative region was reported a decade ago[35]. MRI-embedded inner bore was proposed as a waveguide to deliver RF power at the service end towards the subject placed inside. Even though it has shown potential in large-coverage excitation[36–38], the low efficiency in power transmission makes it impractical to clinical or research use. Also, the $B_1^+$ homogeneity reported in early TW MRI studies was not satisfying, which is due to unmatched phased-velocity.

The coaxial waveguide technique was later introduced to enhance power transmission through wave impedance match[39], and multimode technique alongside with complex $B_1^+$ control methods have been proposed to improve $B_1^+$ homogeneity[40]. However, complex hardware configuration and additional requirement for multi-channel $B_1^+$ control systems make radiative TW excitation technique inaccessible.

Meanwhile, passive components including local resonator array[41] and dielectric materials[42] have also been proposed to enhance local $B_1^+$ in TW MRI. These studies all suggest that TW excitation has certain potential in producing uniform $B_1^+$ under single-channel-transmission mode.

In this study, we present a approach to simultaneously enhance $B_1^+$ homogeneity and power transmission efficiency through structuring a subwavelength dielectric waveguide insert. Principles of manipulating wave behaviors including, i.e., $TE_{11}$-to-$TM_{11}$ mode conversion, power focusing, wave impedance match, and phase velocity match, have been investigated to explore the potential of a well-controlled TW excitation system in addressing challenges of transmit efficiency as well as $B_1^+$ homogeneity.

## Results

### $TE_{11}$-to-$TM_{11}$ mode conversion in TW MRI

In MRI, nuclei spins are excited exclusively by circular-polarized component of the transverse magnetic field. Therefore, transverse magnetic (TM) modes are preferred for efficient TW excitation (see Supplement Fig. 1a). However, the low-order $TM_{11}$ mode, which is favorable for uniform excitation, barely propagates in the embedded circular waveguide of human UHF MRI systems due to the cut-off limit (Supplement Fig. 1b). As shown in Fig. 1, the embedded circular waveguide consists of inner bores of the cryostat (~900 mm in diameter) and gradient coil (~600 mm in diameter). Although the cut-off frequency can be lowered through adding high dielectric fillings[40], it is impractical to propagate $TM_{11}$ mode solely. Additional modes (e.g., $TM_{01}$, $TE_{01}$ and $TE_{21}$) with lower cut-off limits can deteriorate the homogeneity of $B_1^+$. To address this problem, circular dielectric waveguide, which is mainly used as optic fibers, was introduced to achieve mode conversion for efficient TW excitation. Its electrical length was chosen to be smaller than half-wavelength to avoid SW effect. To leave space for imaging subject, dielectric waveguide was designed as a hollow cylinder. Because of zero cut-off frequency, the dominant mode $HE_{11}$ of circular dielectric waveguide can be excited solely inside MRI-embedded metallic circular waveguide[43]. The hybrid mode $HE_{11}$ can be considered as a combination of two major orthogonal modes: $TE_{11}$ and $TM_{11}$. Therefore, the procedure of feeding the dielectric waveguide through metallic waveguide is equivalent to the $TE_{11}$-to-$TM_{11}$ mode conversion.

As shown in Fig. 2f–h, high-index materials from subwavelength dielectric waveguide brings discontinuity in the metallic waveguide, and original wave propagation phase is distorted nonlinearly in local regions. From the perspective of local wave vector, it is refracted by high-index materials as illustrated by Poynting vector $\vec{S}$ shown in Fig. 2i. The refraction angle varies around the hollow dielectric waveguide at subwavelength scale, and there is a nearly 90° refraction in the incident plane.

The deviation of local wave vectors from original propagation direction tilts the transverse electric field vector $E_{r,\theta}$ of $TE_{11}$ mode to be the longitudinal vector $E_z$. According to their symmetry relation, $TM_{11}$ mode can be converted from $TE_{11}$ mode through a 90° flip. The residual $TE_{11}$ mode together with the converted $TM_{11}$ mode constitute the hybrid mode $HE_{11}$, which can be intrinsically carried by the circular dielectric waveguide.

The efficiency of $TE_{11}$-to-$TM_{11}$ mode conversion can be evaluated through quantifying the proportion of residual $TE_{11}$ mode in the hybrid mode $HE_{11}$. Single-mode circular waveguide was used as a filter to measure the residual power of $TE_{11}$ mode through numerical simulations. As shown in Fig. 3b, c, the residual $TE_{11}$ power can be as low as 30% at 297 MHz, suggesting mode conversion efficiency can reach up to 70%. The residual $TE_{11}$ power varies nonlinearly with the frequency, the dielectric constant, and the thickness of hollow cylinder. In addition, its bandwidth becomes broader at higher working frequencies.

### Power focusing

Besides providing larger proportion of transverse magnetic field, the efficiency of producing $B_1^+$ for nuclei spin excitation can be further improved through reducing local wave impedance $Z$ and focusing power in the target region through structuring subwavelength dielectric materials, since the region of interest in MRI only fills a small volume portion within the entire metallic waveguide. In a previous study, distilled water tubes with high dielectric constant were used as waveguide fillings to improve TW MRI efficiency[40]. However, such dielectric tubes were placed in the vicinity of waveguide metallic wall, thus unable to achieve power focusing.

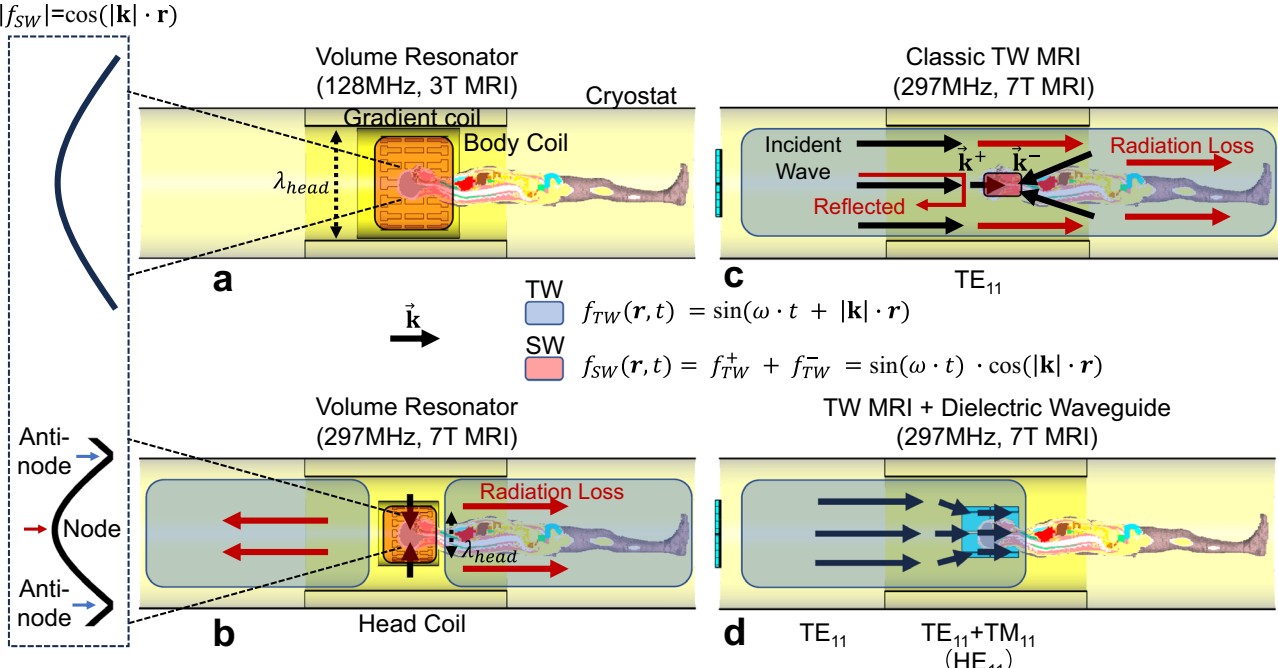

**Fig. 1 | Schematic diagrams of RF transmission systems at 3 T and 7 T for human head MRI.** Body coil (volume resonator) has been widely equipped in 3 T MRI systems for whole-body standing wave (SW) excitation. The wavelength $\lambda_{head}$ (at 128 MHz) in biological tissues is larger than the human head. The magnitude variation $\cos(|\mathbf{k}| \cdot \mathbf{r})$ of SW is trivial in the human head (**a**). In comparison, local birdcage coil (volume resonator) has been industry standard for human head imaging at 7 T. Since the wavelength $\lambda_{head}$ (at 297 MHz) in biological tissues approaches human head dimension, node and antinode of SW appear in the human head. The human body and the inner metallic surface of the MRI bore (the waveguide) become electrically large, so the volume resonator can radiate power inside, and the waveguide carries $TE_{11}$ mode travelling wave (TW) at 297 MHz (**b**). However, the human body introduces discontinuity in wave impedance and phase velocity in the waveguide, leading to reflected power and secondary SW (**c**). To this end, dielectric waveguide has been proposed to achieve efficient TW excitation through $TE_{11}$-to-$TM_{11}$ mode conversion, power focusing, wave impedance match, as well as phase velocity match (**d**).

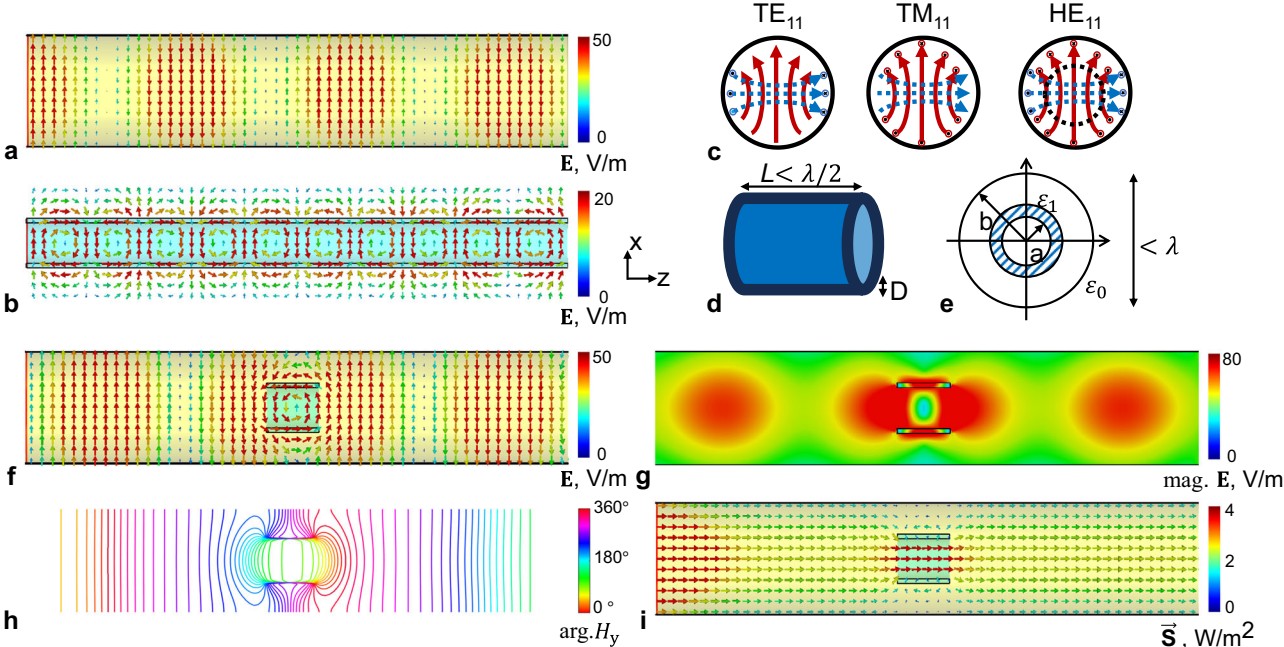

**Fig. 2 | The mechanism of proposed $TE_{11}$-to-$TM_{11}$ mode conversion through subwavelength dielectric waveguide.** The electric field in the XZ plane of the dominant $TE_{11}$ mode in a circular metallic waveguide (**a**) and the $HE_{11}$ mode in a hollow dielectric waveguide (**b**); The schematic field plot of $TE_{11}$, $TM_{11}$ modes and hybrid $HE_{11}$ mode (**c**). The red solid vector plot indicates the electric field, while the blue dashed vector plot indicates the magnetic field; the schematic drawing of the subwavelength dielectric waveguide (**d**) and its transverse plot in a metallic circular waveguide (**e**). The $TE_{11}$-to-$TM_{11}$ mode conversion by subwavelength dielectric waveguide with minimized interference to wave propagation in the metallic waveguide, indicated by the electric field distribution (**f**, **g**). The waveguide discontinuity introduced by the subwavelength waveguide which is indicated by distortion in phase propagation (**h**) as well as Poynting vectors (**i**).

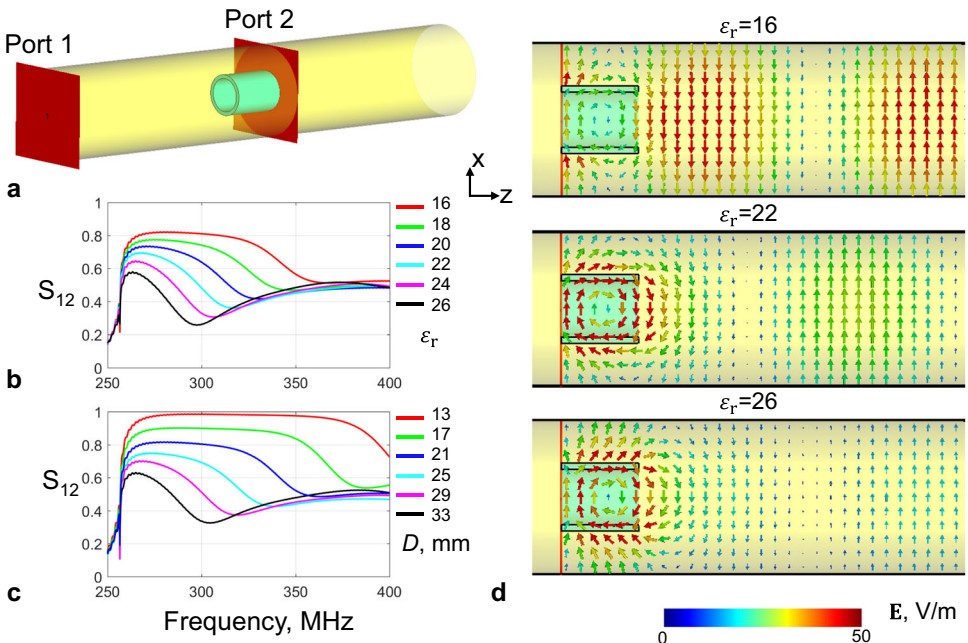

**Fig. 3 | The efficiency of TE$_{11}$-to-TM$_{11}$ mode conversion.** Measured by transmission efficiency S$_{12}$ of a two-port waveguide system. The empty part of the metallic waveguide acts as a filter in which only TE$_{11}$ mode can propagate (**a**). The dependency of mode conversion efficiency with relative permittivity $\varepsilon_r$ (**b**) as well as wall thickness D of the dielectric waveguide (**c**). The electric field distributions in the metallic waveguide are reshaped by dielectric waveguide insert (**d**). The TE$_{11}$-to-TM$_{11}$ mode conversion efficiency is indicated from the residual electric field (TE$_{11}$) in the remaining part of the metallic waveguide without dielectric fillings, where TM$_{11}$ mode cannot propagate. The wall thicknesses in (**b, d**) are kept the same as 28 mm, while the relative permittivity $\varepsilon_r$ in **c** is kept the same as 21.

The power focusing effect of dielectric waveguide insert has been investigated in this study. As shown in Fig. 4, the power flow density $|\vec{\mathbf{S}}|$ in an empty metallic waveguide is uniformly distributed along the wave propagation direction and in the transverse cross section. Since power flow density $|\vec{\mathbf{S}}| = \frac{\mathbf{E}_{peak}^2}{2\eta}$ scales with the inversion of intrinsic impedance $\eta = \sqrt{\frac{\mu}{\varepsilon}}$, dielectric materials can be structured to enhance local power flow in target region of interest. The effect of dielectric waveguide in local power focusing was evaluated through numerical simulations of Poynting vector $\vec{\mathbf{S}}$. As shown in Fig. 4b, power flow density inside dielectric waveguide is apparently enhanced compared to surrounding areas of the metallic waveguide. According to the normalized power flow density distribution across center transverse plane of the dielectric waveguide, the power focusing effect varies nonlinearly with the radius of the dielectric waveguide. Due to the large discontinuity of the normal E field away from dielectric boundaries, the power density distribution can even peak outside of the dielectric waveguide. Therefore, the radius of dielectric waveguide should be carefully designed to achieve desired power focusing.

In addition to efficient excitation, deliberate power focusing can also achieve targeted MRI with minimized aliasing ghosts[44]. Due to the nature of radiative TW propagation, excited area usually extends beyond preset imaging FOV, which causes aliasing effect if the phase encoding direction is set along the wave propagation direction. By incorporating a dielectric waveguide for power focusing, efficient excitation is achieved specifically in the head and neck regions, as opposed to the broader excitation of a classic MRI waveguide (see below).

**Wave impedance matching**

The existence of biological subject introduces discontinuity in wave impedance, therefore it leads to an unmatched condition in power transmission. The coaxial waveguide design has been introduced to achieve wave impedance match[39], and up to 40% of total stimulated power can be transferred to the load. However, the giant conductive metal insert required can increase burning risks caused by eddy current during MRI scans. Here, we demonstrate that the dielectric waveguide structure can also be used for wave impedance match. As shown in Fig. 5, the dielectric waveguide along with a lossy dielectric cylinder ($\varepsilon_r$ = 68; $\sigma$ = 0.5 S/m) as the load were placed inside a metallic circular waveguide. The reflection coefficient S$_{11}$ of the driving waveguide port, as well as power loss compositions, were used to evaluate wave impedance match conditions. The wave impedance match condition varies with the length of the dielectric waveguide. In addition to minimizing power reflection, the dielectric waveguide has demonstrated the ability in reducing radiation loss. As shown in Fig. 5c, the power delivered to the load was improved from 15% to 70%.

**Phase velocity matching**

Besides the transmission efficiency, TW MRI also have shown disadvantages in $\mathbf{B}_1^+$ homogeneity for nuclei spin excitation[45]. According to the boundary condition of conductive media, the propagation constant $\beta \approx \omega\sqrt{\mu\varepsilon}$ inside and outside lossy tissues are discontinuous. Due to the large dielectric constant (~60) of biological tissues, phase velocities $\upsilon_p = \frac{\omega}{\beta}$ between electrically large human bodies and air fillings of metallic waveguide are severely unmatched. As a result, incoherent wavefront formed at the air-tissue boundary triggers opposing propagating waves and results in the formation of SW within electrically large human body, as shown in Fig. 1c. The SW effect is clearly indicated from the phase map, characterized by a rapid 180° phase alternation and interleaved with constant phase regions. The dark spot in $\mathbf{B}_1^+$ overlaps within the region of rapid phase changes.

To solve this problem, the dielectric waveguide with boundary condition of continuous tangential field components was used to achieve phase velocity match. The continuous tangential field assures the same propagation constant at air-dielectric boundaries. Through increasing the permittivity (higher than that of the air) of the dielectric waveguide, the propagation constant in surrounding areas of the

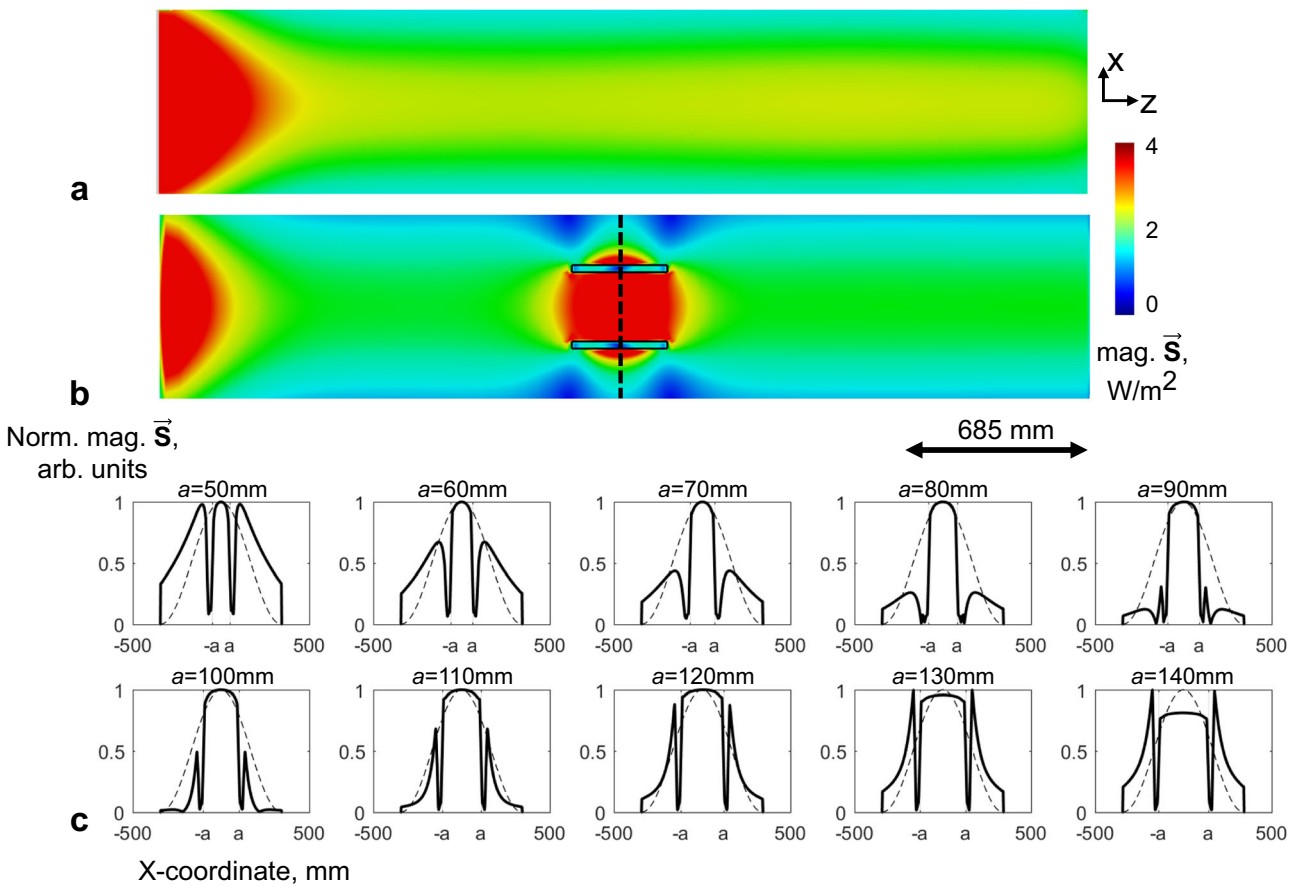

**Fig. 4 | Power focusing effect of the subwavelength dielectric waveguide in a metallic waveguide.** Energy flow density $\vec{S}$ in an empty circular metallic waveguide (**a**) and in a discontinuous metallic waveguide with subwavelength dielectric waveguide insert (**b**). The power focusing effect is illustrated through a 1D plot of normalized $\vec{S}$ magnitude in the transverse plane (**c**). The black sold lines in (**c**) indicate 1D plot sampled from (**b**), while the black dashed lines indicate 1D plot sampled from (**a**).

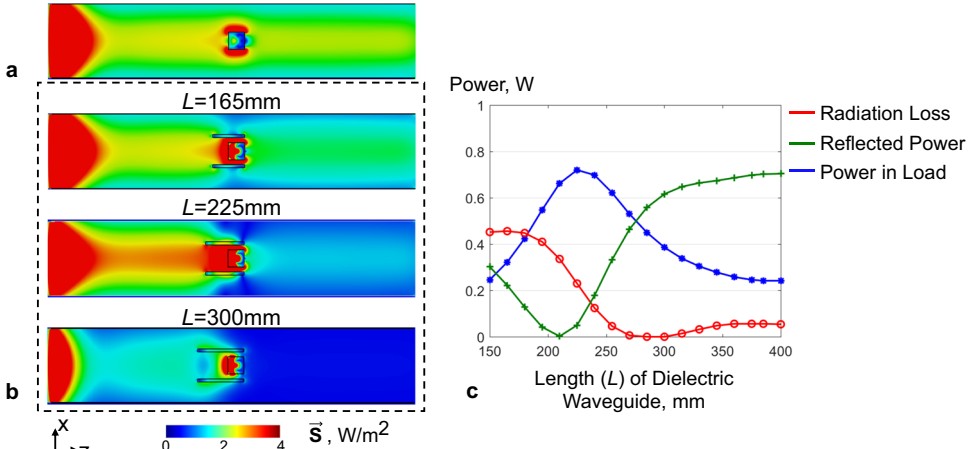

**Fig. 5 | Wave impedance match with dielectric waveguide.** Energy flow density $\vec{S}$ in a circular metallic waveguide loaded with a lossy dielectric cylinder (160 mm in diameter, 150 mm in length; $\varepsilon_r$ = 68, $\sigma$=0.5 S/m) (**a**), and in a circular metallic waveguide with both load and dielectric waveguide inserts for wave impedance match (**b**). The power dissipated in the load, power reflected and the radiation power loss vary with the length of the dielectric waveguide (**c**).

human body can be increased to approach the value in lossy biological tissues. As a result, the well-matched condition of phase velocities can be achieved, and the SW effect inside the dielectric rods was alleviated to obtain uniform field distribution, as shown in Fig. 6. In addition, the propagation constant $\beta$ of the dielectric waveguide varies with its radius, dielectric constant, and wall thickness. Therefore, well-matched condition can be achieved in dielectric waveguides with different structural properties.

## Human head TW MRI at 7T
Human MRI at 7 T has suffered from $\mathbf{B}_1^+$ inhomogeneity for decades. Even though it can be alleviated with multi-channel transmission

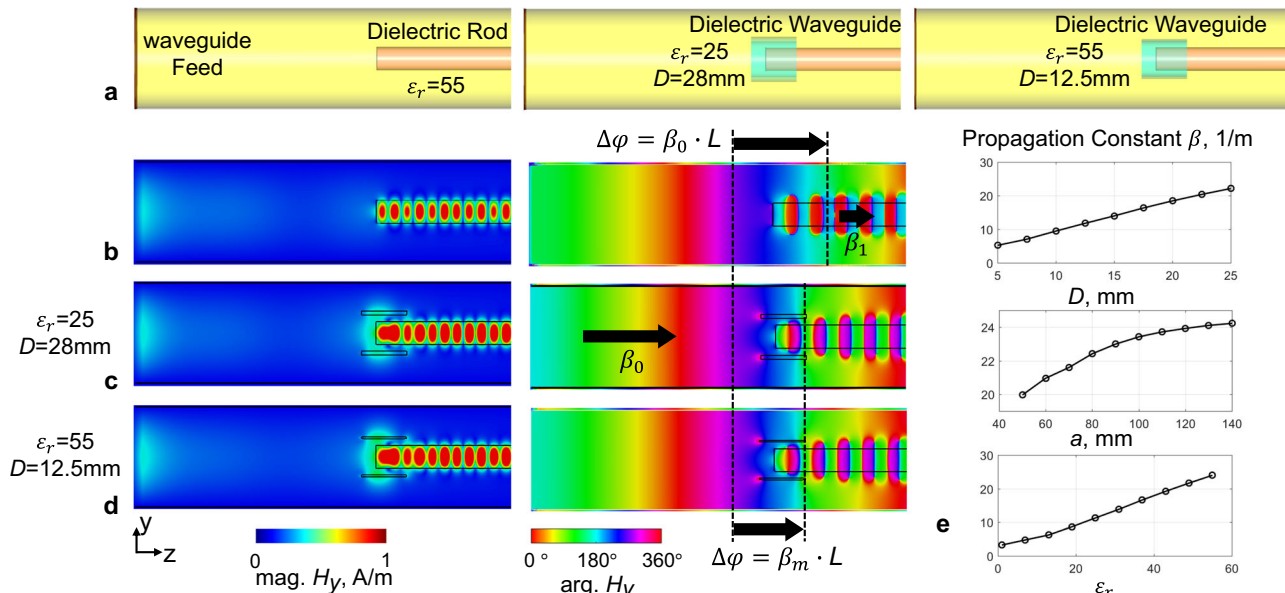

**Fig. 6 | Phase velocity match with dielectric waveguide.** Lossless dielectric rod (160 mm in diameter, $\varepsilon_r = 55$) was inserted inside a circular metallic waveguide to demonstrate standing wave (SW) effect due to phase velocity mismatch. Besides, two types of dielectric waveguide inserts (28 mm thickness, $\varepsilon_r = 25$; 12.5 mm thickness, $\varepsilon_r = 55$) were used to achieve phase velocity match for the load (**a**). The magnitude and phase of $H_y$ of unmatched dielectric rod (**b**), phase velocity matched dielectric rods with two types of dielectric waveguide insert (**c**, **d**). Propagation constant $\beta$ varies with the thickness $D$, radius $a$, and dielectric constant $\varepsilon_r$ of the dielectric waveguide (**e**).

system, it has still been an unmet need for clinical use, where only single-channel-transmission mode is allowed due to safety concerns. The electrical length of the human body at the operation frequency (297 MHz) determines that the classic quasi-static excitation method is difficult to produce uniform $\mathbf{B}_1^+$ over the entire human body. Even for the human head, the canonical birdcage resonator produces center-brightening effect as well as dark voids (e.g., in temporal lobes) due to the SW effect at 7 T (see below). As discussed above, TW excitation is a promising solution to produce uniform $\mathbf{B}_1^+$ by nature. However, current TW methods suffer from low transmission efficiency as well as secondary SW, which prevent its further applications.

To overcome the problem, we proposed a modified TW MRI solution based on subwavelength dielectric waveguide, which is compatible with single-channel transmission (the clinical mode). A human 7 T MRI scanner with the bore diameter of 685 mm was chosen to examine its efficacy. Its performance in MRI nuclei spin excitation was compared with a product birdcage resonator which has been routinely used for human head imaging at 7 T under the clinical mode. Since multi-channel transmission systems are only available for research purposes, the state-of-the-art phased array resonator coils were not included for comparison in this study.

As shown in Fig. 7a–c, multiple dielectric cubes (6 × 2) were placed around a cylindrical surface to constitute a hollow dielectric waveguide. Compared to the classic design with continuous structures, cubic array design provides extra flexibility for fine adjustment in practical use. For example, the gap between neighboring cubes along the longitudinal direction can fine adjust the equivalent dielectric waveguide length for wave impedance match; the inner radius of hollow dielectric waveguide can be easily altered to fit the human subject; and through replacing the supporting cylindrical structure, it can be conveniently configured for power focusing purposes. Due to discretized structure of dielectric waveguide, its equivalent dielectric constant is lowered. As shown in Fig. 7d, the dielectric cubic array of the dielectric constant $\varepsilon_r = 52$ has shown similar effect in RF field manipulation with the dielectric cylinder of the dielectric constant

$\varepsilon_r = 21$. Its effect on transverse magnetic field $H_{xy}$ enhancement can be observed in Fig. 7g.

A circular patch antenna (driven by port 1 and 2) was used in this study to feed the metallic waveguide. Two ports in patch antenna were quad-driven to produce circular-polarized magnetic field in order to maximize $\mathbf{B}_1^+$. To fine-tune the proposed dielectric waveguide in practice, a sniffer magnetic-loop probe (driven by port 3 in simulation) was used and is shown in Fig. 7e. The power transmission efficiency was evaluated through measuring $S_{31}$ and $S_{32}$, and the difference between $S_{31}$ and $S_{32}$ indicates that two linear polarized fields with high orthogonality were delivered (see Fig. 7h). The peak power transfer efficiency was 10 dB, higher than the off-resonance region in both simulation and experiment measurements, indicating effective enhancement of transverse magnetic field by using the dielectric waveguide.

Detailed information about optimized dielectric waveguide structure is presented in the Method section. Its performances in $\mathbf{B}_1^+$ enhancement as well as phase velocity match with loaded human body were evaluated and compared to a canonical birdcage resonator through numerical simulations and MRI experiments at 7 T. As shown in Fig. 8, the modified TW method has demonstrated a reduction of 22.3% in $\mathbf{B}_1^+$ inhomogeneity over brain region in the chosen sagittal plane, compared to the classic TW method. Through phase-velocity match, the prominent center-brightening effect and dark voids in temporal lobes in birdcage resonators were greatly inhibited. Moreover, modified TW method extends signal excitation towards the neck region, which is particularly useful in head and neck imaging such as arterial spin labelling[46]. Compared to the birdcage resonator, the modified TW method has shown 22.4% and 23.2% reduction in $\mathbf{B}_1^+$ inhomogeneity over brain regions in the chosen coronal plane and transverse plane (see Fig. 9). The modified TW method has achieved reductions of 8.7% and 21.9% in $\mathbf{B}_1^+$ inhomogeneity across the entire brain compared to the classic TW method and the birdcage resonator, respectively (see Table 1).

The modified TW method with dielectric waveguide has shown comparable $\mathbf{B}_1^+$ efficiency with the birdcage resonator (0.35 uT/$\sqrt{W}$

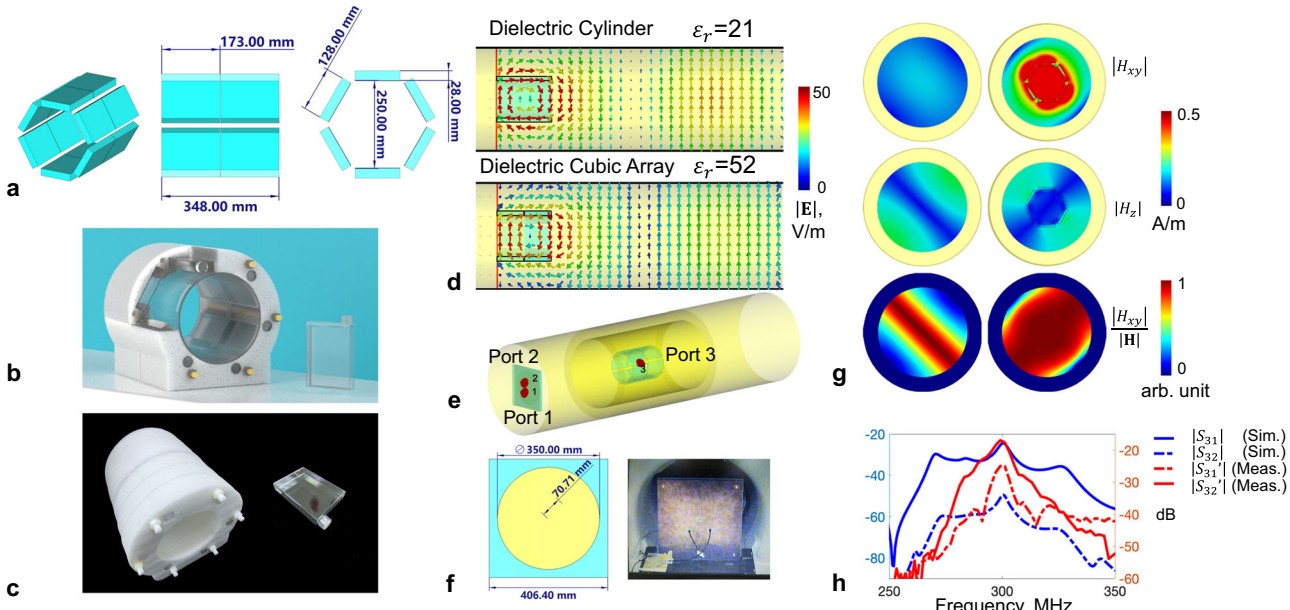

**Fig. 7 | Subwavelength dielectric waveguide for human head TW MRI at 7 T.** 6 × 2 dielectric cubes were used to constitute the hollow dielectric waveguide to provide higher structural flexibility (**a**). The EPE foam was used to attach cubic array along a cylindrical surface (**b**, **c**). The electric field distribution in a dielectric cylinder ($\varepsilon_r = 21$) inserted into the circular metallic waveguide, and the electric field distribution in a dielectric cubic array ($\varepsilon_r = 52$) inserted into the circular metallic waveguide (**d**). The modified single-channel transmission TW waveguide system with dielectric cubic array insert. It was fed with a classic two-port patch antenna, interfaced with a 90° quad-hybrid to provide circular polarization. **e** The schematic and the photograph of the quad-driven feeding antenna (**f**). The transverse magnetic field component $H_{xy}$ vs. the longitudinal magnetic field component $H_z$ in the empty circular metallic waveguide (**g**, left) and in a dielectric waveguide inserted into the circular metallic waveguide (**g**, right). The transmission coefficient between the magnetic probe placed inside dielectric waveguide and the feeding antenna (**h**), calculated and measured in numerical simulations ($S_{31}$, $S_{32}$) and in MRI experiments ($S_{31}'$, $S_{32}'$), respectively.

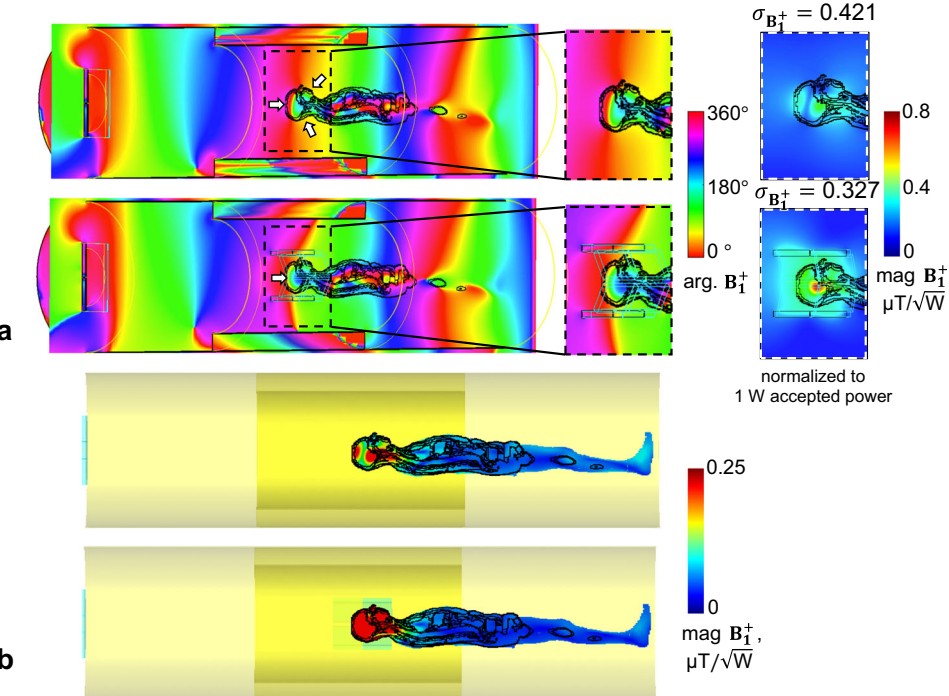

**Fig. 8 | Phase velocity match for human head MRI in a metallic circular waveguide.** The $\mathbf{B}_1^+$ phase distribution inside the waveguides and $\mathbf{B}_1^+$ magnitude (with inhomogeneity quantified as normalized mean square error $\sigma_{\mathbf{B}_1^+}$ across the entire brain) are shown in **a**. The distribution of $\mathbf{B}_1^+$ magnitude within the human subject in an MRI-embedded waveguide is displayed in **b**, with the color range re-adjusted to [0, 0.25] to better showcase the efficacy of power focusing within the human head region.

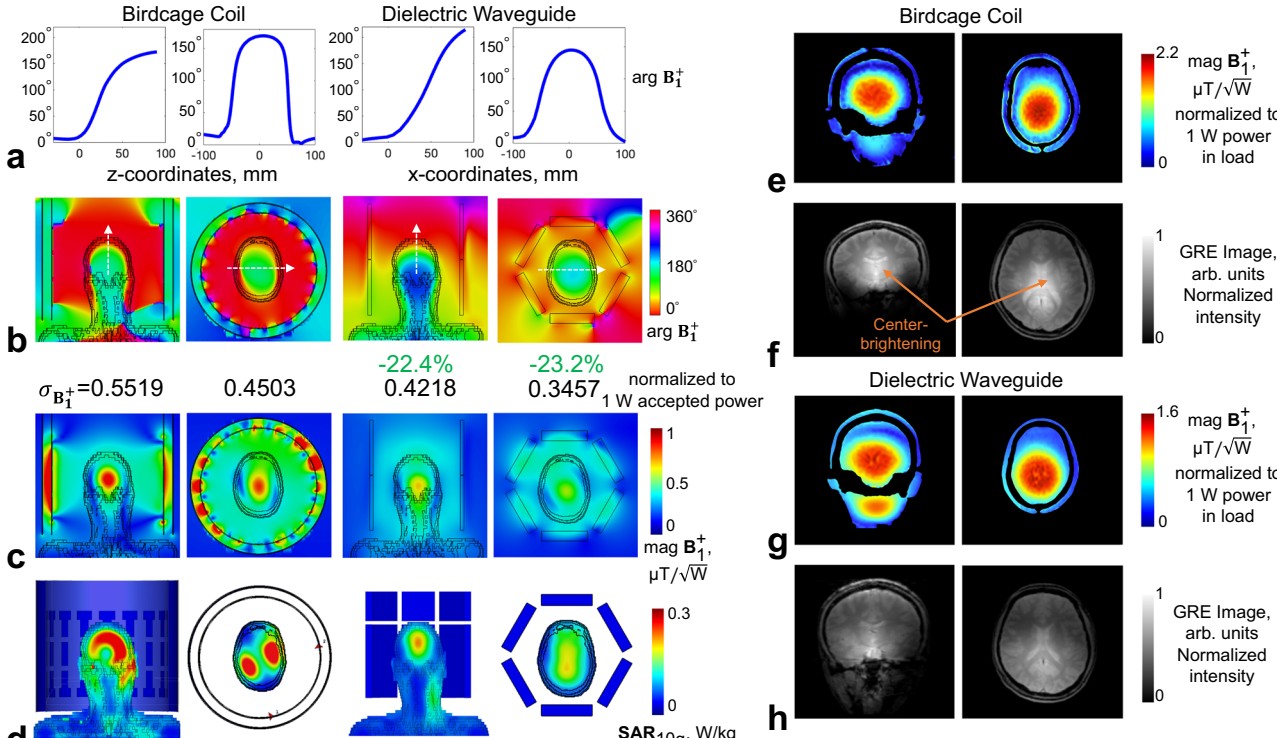

**Fig. 9 | Human brain MRI by using a birdcage coil vs. modified TW system at 7 T.** Both approaches were quad-driven under single-channel transmission mode. 1D phase evolution **a** along white dashed lines indicated in (b). Phase maps of $\mathbf{B}_1^+$ in a coronal slice (left) and in a transverse slice (right) (**b**). B1+ magnitude in a coronal slice (left) and in a transverse slice (right). Normalized mean square error $\sigma_{\mathbf{B}_1^+}$ were calculated over each brain slice (**c**) and 10-average SAR (**d**). MRI experiments results of quantitative $\mathbf{B}_1^+$ maps acquired from an anthropomorphic head phantom (**e**, **g**), and proton-density weighted GRE images acquired from a human subject in vivo (**f**, **h**).

vs. 0.428 uT/$\sqrt{W}$ in average $\mathbf{B}_1^+$ magnitude) over the entire human brain (see Fig. 9). Due to characteristics of reactive and radiative regions, modified TW method has shown much lower SAR compared to the birdcage resonator (0.331 W/kg vs. 0.593 W/kg in maximum 10g-average local SAR). As a result, for the proposed TW method, $\mathbf{B}_1^+$ efficiency normalized by SAR is 10.8% higher than that of the birdcage resonator. It should be noted that, the resistive power loss from transmitters is underestimated in full-wave numerical simulations.

In MRI experiments, quantitative $\mathbf{B}_1^+$ maps were acquired using the AFI (actual-flip angle) sequence[47]. Since reliable quantitative SAR mapping method in MRI is still missing, the power loss in load was used to normalize the measured $\mathbf{B}_1^+$ map in this study (normalized to 1 W power in load). Such normalization can be achieved through measuring the quality factor ratio ($Q_{ratio}$: $\frac{Q_{unloaded}}{Q_{loaded}} = \frac{P_{Tx-loss} + P_{load-loss}}{P_{Tx-loss}}$), in which $P_{Tx-loss}$ is power loss from the transmitter (including waveguide in the TW method), and $P_{load-loss}$ is power loss in load, while $Q_{unloaded}$ and $Q_{loaded}$ can be measured using the standard double-probe method[48]. The patch antenna for TW excitation was placed at the service end of the MRI scanner bore for quality factor measurement. The normalization factor $Norm._{1W\text{-}load}$ was calculated as: $\frac{1}{(1 - \frac{1}{Q_{ratio}})} = \frac{P_{Tx-loss} + P_{load-loss}}{P_{load-loss}}$.

Then, the $\mathbf{B}_1^+$ normalized to 1 W power in load can be obtained as: $\mathbf{B}_1^+{}_{1W-load} = \mathbf{B}_1^+{}_{1W-power} \cdot Norm._{1W\text{-}load}$. In this study, the $Q_{ratio}$ of the patch antenna (waveguide side) and the birdcage resonator were measured as 1.03 and 2.5 respectively. The $Norm._{1W\text{-}load}$ for TW was 4.5 times larger than that of SW. Therefore, the TW showed similar transmit efficiency in $\mathbf{B}_1^+{}_{1W-load}$. It agrees well with numerical simulation results in Fig. 9c, where the resistive power loss is underestimated.

Furthermore, the dielectric-waveguide-based TW method was also compared with state-of-the-art implementations of TW MRI[39,40] in single-channel-transmission mode. As shown in supplementary Fig. 2, the proposed TW method shows improved performance in SAR normalized $\mathbf{B}_1^+$ efficiency and homogeneity, which can be attributed to its key advantages in effective $TE_{11}$-to-$TM_{11}$ mode conversion, power focus, wave impedance match, and phase velocity match.

## Discussion

We present a solution for improved TW MRI excitation in electrically large human subjects with subwavelength dielectric waveguide. Its characteristics in, i.e., $TE_{11}$-to-$TM_{11}$ mode conversion below the cut-off limit, power focusing, wave impedance matching, and phase velocity matching have been investigated through numerical simulations and imaging experiments. With advantages in transverse magnetic field enhancement and maximized power transmission, the TW MRI efficiency in producing $\mathbf{B}_1^+$ was improved by 114% over the entire human brain compared to classic TW method. SAR normalized efficiency (mag.$\mathbf{B}_1^+/\sqrt{\max. SAR10g}$) of modified TW method was 10.8% higher than traditional canonical birdcage resonator. Through phase velocity matched TW excitation, $\mathbf{B}_1^+$ inhomogeneity was reduced by 21.9% compared to the birdcage resonator. Such benefits were also validated through MRI experiments with both quantitative $\mathbf{B}_1^+$ mapping and anatomical imaging on a commercial 7 T human MRI scanner. The well-known center-brightening phenomenon has been effectively inhibited, according to the GRE images shown in Fig. 9f, h. The $\mathbf{B}_1^+$ distribution quantitatively measured on the head phantom (shown in Fig. 9e, g) was however not very uniform, because the relative permittivity (78) of the phantom material is higher than that of the human brain (40-60), which led to sub-optimal wave impedance match. Nonetheless, the $\mathbf{B}_1^+$ distribution along x-direction was more uniform in TW transmission.

**Table 1 | Numerically simulated results of the $B_1^+$ and local SAR efficiencies for different transmission methods**

| Transmission methods | MEAN. $\{mag.B_1^+\}$ (µT/$\sqrt{W}$) | Norm. $\sigma_{B_1^+}$ (µT/$\sqrt{W}$) | Max. SAR10g (W/kg) | MEAN. $\{mag.B_1^+\}/\sqrt{max.SAR10g}$ (µT/$\sqrt{W}$/kg) |
|---|---|---|---|---|
| Birdcage | 0.428 | 0.4836 | 0.593 | 0.556 |
| Classic-TW | 0.166 | 0.4199 (−13.2%) | 0.125 | 0.469(−15.6%) |
| Dielectric-Waveguide | 0.3545 | 0.3777 (−21.9%) | 0.331 | 0.616 (+10.8%) |

All values were taken from human brain regions.

In this study, measured $B_1^+$ maps were normalized to 1 W power in load ($B_{1\ 1W-load}^+$) instead of 1 W input power ($B_{1\ 1W-power}^+$). Because the birdcage resonator (SW) and radiative method (TW) work fundamentally differently in power delivering, the classic $B_1^+$ normalization method with reference to 1 W input power ($B_{1\ 1W-power}^+$) is insufficient to account for such disparity. The nature of radiative transmission, i.e., lower filling factors and higher proportions of power loss from transmitters, leads to reductions in both of $B_1^+$ and SAR. Therefore, SAR normalized transmit efficiency (mag. $B_1^+$/$\sqrt{max.SAR}$) is preferred as a more reasonable quantitative assessment[10,49]. However, reliable quantitative methods for SAR mapping in MRI are indeed still limited. Although MRI Thermometry can be used as an alternative method to indirectly evaluate SAR distribution in saline phantoms, conventionally equipped 8 kW power amplifier paired with local resonators usually cannot guarantee adequate amount of power delivered to the load with TW transmission. Therefore, the temperate change may not be significant enough to be captured by MRI Thermometry. Consequently, power loss in load was used to normalize $B_1^+$ in this study. Such normalization was achieved through measuring the quality factor ratio with the standard double-probe method. According to the normalized $B_{1\ 1W-load}^+$ maps showed in Fig. 9e, g, the modified TW method has shown comparable transmit efficiency with the industry-standard birdcage resonator.

According to the low $Q_{ratio}$ of the TW method measured in this study, the power loss from the transmitter (including waveguide) was much more dominant compared to the power loss in load. It may attribute to the imperfect waveguide structure and materials which consist of cryostat inner bore and gradient shied (metallic mesh). As a result, 484 V in transmit voltage would be required in order to ideally achieve 90° spin excitation in an anthropomorphic head phantom. Limited by the peak power (8 kW) of the RF amplifier equipped on the 7 T MRI scanner which was designed to be paired with local resonators, only up to 120° refocusing pulse can be achieved. Therefore, an upgraded power amplifier is preferred to achieve strict 180° refocusing pulses.

The low filling factor nature of the TW method indeed makes it more comparable to body coil transmitters than local resonators in MRI, as shown in Fig. 1. For example, 1.5 T MRI body coils are typically equipped with 15–20 kW power amplifiers, while 3 T body coils are usually paired with 35 kW power amplifiers, which can be attributed to elevated power loss during RF power transmission as well as dielectric loss in loadings[50]. It is worth noting that body coils are typically employed for whole-body excitation, encompassing the head and other body organs. However, in the context of our current study, the utilization of 20–35 kW power amplifiers is deemed sufficient to achieve 180° refocusing pulses, thereby enabling an efficient implementation of the TW method at 7 T. Additionally, by replacing existing embedded waveguide structures with low-loss waveguides, we can further reduce the power requirements of the amplifiers, enhancing the overall energy efficiency of the MRI system.

## Methods
### Numerical simulations
The dielectric waveguide and its effect in TW MRI at 7 T were simulated and evaluated using full-wave numerical simulation software (CST, Dassault Systèmes, France), operating with the center frequency at 297 MHz. Time-domain solver was used to calculate electromagnetic fields and scatter parameters. Two types of metallic waveguides were simulated in this study: (1) as shown in Fig. 1, a circular waveguide with the diameter of 685 mm to investigate principles of hollow dielectric waveguide in $TE_{11}$-to-$TM_{11}$ mode conversion, power focusing, wave impedance match as well as phase velocity match, and (2) as shown in Fig. 7e, a circular waveguide with stepped-diameter[36] to mimic the realistic MRI-embedded waveguide, which consists of a copper cylindrical shield representing the cryostat (900 mm in diameter, 3360 mm in length), alongside with a narrow copper cylindrical shield in the center representing the RF shield (685 mm in diameter, 1220 mm in length).

A patch antenna consists of a circular copper patch (350 mm in diameter) and a ground shield interleaved with an acrylic slab was modelled to feed the waveguide (Fig. 7f). It was driven with two discrete feeding ports in the quadrature mode to maximize $B_1^+$ field. Each feeding port is located 70.7 mm distance away from the circular patch center for 50Ω matching. The feed antenna was placed at one end of the waveguide.

The cubic array dielectric waveguide was modelled with materials of relative permittivity of 52 and conductivity of 5.55e−6 S/m. As shown in Fig. 7a–c, all dielectric cubes were arranged on a cylindrical surface, with cylinder diameter and cube thickness optimized (173 mm in length, 128 mm in width and 28 mm in thickness, equally distributed with diameter of 250 mm) in order to achieve maximum $TE_{11}$-to-$TM_{11}$ conversion at 297 MHz.

A classic 16-rung band-pass birdcage coil was modelled for comparison. All rungs and end-rings were arranged on a cylindrical surface (335 mm in diameter), and the cylindrical RF shield (390 mm in diameter) covers the entire birdcage coil. Each rung (240 mm in length and 25 mm in width) was divided into 3 segments connected with 2 lumped capacitors. Human model Gustav was imported for full-wave numerical simulation.

### Subwavelength dielectric waveguide
The cubic array dielectric waveguide was arranged on a cylindrical shape for human head imaging. Each cube was manufactured as a polycarbonate container filled with water-sucrose solution. Distilled water was used to achieve low conductivity, and water/sucrose mass ratio of 100:54 was used in order to achieve optimal power transmission as indicated in $S_{31}$' at 297 MHz (measured by a portable VNA) – the relative permittivity and conductivity were measured as 70 and 0.02 S/m respectively. Its frequency-selective response can be measured by the power transmission coefficient between a waveguide feed antenna and a magnetic field probe located inside the dielectric cylinder (as shown in Fig. 7e).

An acrylic cylinder (250 mm outer diameter, 5 mm thickness and 390 mm in length) was used to support the dielectric cubic array. In total, 12 cubic polycarbonate containers (each with 175 mm in length, 130 mm in width, 30 mm in thickness and 1 mm wall thickness) were filled with sucrose-water solution and distributed equally outside the acrylic cylinder. EPE (expanded polyethylene) foam with minimized field interference was used to assist in attaching the cubic array closely to the outer surface of acrylic cylinder (Fig. 7b, c).

## MRI experiment

All MRI experiments were conducted on a 7 T human MRI scanner (MAGNETOM 7 T, Siemens Healthcare, Erlangen, Germany) operating in the clinical mode (single-channel transmission). A circular patch antenna was constructed to deliver electromagnetic waves in the waveguide. It consists of a circular copper patch (350 mm in diameter) and a ground sheet interleaved with two acrylic slabs. The gap distance between slabs were adjustable for fine tuning. The patch antenna, operating in the transceiver mode through connecting to a T/R switch, was driven with two feeding ports interfaced with a 90° quad-hybrid. Each feeding port is located 70.7 mm away from the circular patch center for 50Ω match. The feed antenna was placed at the service end of the bore. The subwavelength dielectric waveguide was placed on the patient table and positioned at isocenter of the MRI magnet as well as of the waveguide. The standard RF coil for clinical human head MRI at 7 T, i.e., Nova 1Tx/32Rx head coil (Nova Medical, MA, US) was used for comparison, and its transceiver mode was also used for fair comparison.

A brain-tissue-mimicking anthropomorphic head phantom[51] was used for imaging. Quantitative $B_1^+$ mapping sequence (actual-flip-angle, AFI)[47] (TR1/TR2: 20 ms/50 ms; TE: 2.53 ms; voxel size: $3.6 \times 1.8 \times 3$ mm$^3$) was used to evaluate the excitation efficiency as well as homogeneity. Reference voltages for driving the patch antenna and the Nova birdcage coil were set to 400 V and 100 V respectively. The quality factor ratio ($Q_{ratio}$: $\frac{Q_{unloaded}}{Q_{loaded}} = \frac{P_{Tx-loss} + P_{load-loss}}{P_{Tx-loss}}$), which was considered in normalizing $B_1^+$ to 1 W power in load for fair comparisons, was measured for both transmitters using the standard double-probe method[48].

In vivo study over a healthy volunteer was conducted with all procedures approved by the Ethics Committee at Zhejiang University (2022-45) and with written informed consents obtained from the subject. AFI-B1 mapping was not considered due to its short RF pulse duty cycle and large flip angle excitation which may raise SAR concerns; instead, GRE T2* images (TR: 1000 ms, TE: 3.54 ms, nominal flip angle: 60; voxel size:$1.5 \times 1.5 \times 2$ mm$^3$, scan time: 1'17") were acquired to qualitative evaluate the excitation homogeneity.

## Data availability

The data that support the findings of this study are available on request from the corresponding authors. The data are not publicly available due to privacy or ethical restrictions.

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

## Acknowledgements

We thank Zhejiang University 7 T Brain Imaging Research Center for helpful technical support. This work was supported in part by STI 2030 - Major Projects 2021ZD0200401 (X.Z.), National Natural Science Foundation of China 52277232 (X.Z.), 52307256 (Y.G.), 52293424 (Y.F. and X.Z.), 81701774 (X.Z.), 61771423 (X.Z.), Postdoctoral Science Foundation of China 2020M681866 (Y.G.), the Fundamental Research Funds for the Central Universities 226-2022-00136 (X.Z.), 226-2023-00125 (X.Z.), XJSJ23009 (Y.G.), Zhejiang Provincial Natural Science Foundation of China LR23E070001 (X.Z.), Key R&D Program of Jiangsu Province BE2022049 (X.Z.), and Key-Area R&D Program of Guangdong Province 2018B030333001 (X.Z.), Proof of Concept Foundation of Xidian University Hangzhou Institute of Technology GNYZ2023YL0404 (Y.G.).

## Author contributions

Y.G. and X.Z. conceptualized the work. Y.G. and T.L. performed electromagnetic simulations, data analysis, and carried out the experiment. Y.G. and X.Z. designed and performed MRI experiments. T.H. and Y.F. contributed to the interpretation of the results. Y.G. and X.Z. wrote the manuscript with input from all authors. W.J. and X.Z. supervised the findings of this work. All authors discussed the results and contributed to writing the manuscript.

## Competing interests

The authors declare no competing interests.
