## [Peer Review File · Nature Communications]

Subwavelength Dielectric Waveguide for Efficient Travelling-Wave Magnetic Resonance ImagingReviewer #1 (Remarks to the Author):

Overall comments: This paper returns to a decade-and-a-half-old concept of traveling wave MRI, in which the cylindrical bore of the MR scanner serves to propagate RF fields required for spin excitation and signal generation. The authors propose a mode-conversion mechanism to improve the performance of traveling wave excitation. While this mechanism is interesting (though not convincingly explained in the paper), the resulting excitation field distributions are not likely to be of much practical use in research or clinical MRI. The authors overstate the problem of inhomogeneous excitation, for which there are already some reasonable mitigation strategies. Meanwhile, the excitations their traveling-wave transmitter does produce are only marginally improved over a simple birdcage coil excitation, which can already be improved upon substantially, in the head at least, with relatively simple coil arrays and RF shimming. Moreover, the results presented here are unlikely to be generalizable beyond the head, given the interactions of RF fields with the complex dielectric structures in the rest of the body. In light of these limitations, I do not believe that the True-Traveling-Wave approach proposed in this paper will have much sway in MRI circles. Potential applications of related technologies in nano-scale photonics and sensors are mentioned, but not elaborated upon, so it is also difficult to assess any broader impact.

Detailed comments:

- Abstract, lines 38-39, and also Introduction, lines 78-79: "Uniform B1+ is key to producing favorable tissue contrast..." I would not actually characterize B1+ uniformity as 'key,' since a wide range of transmit coils and inhomogeneity mitigation techniques exist, and since some applications do not even require uniform contrast. Uniformity is 'helpful,' perhaps, but not key.
- Abstract, lines 39-40, and Introduction, lines 83-85: "While the standing wave...can full meet such requirements." RF excitation in most cases does not actually involve a standing wave – just a quasi-static field distribution. At high field strength and high operating frequency, standing wave effects can contribute, but field inhomogeneity can result at least as much from coil geometry and symmetry as from actual interference of forward and reflected waves. This is partly a matter of semantics, but a paper on electrodynamic effects should be precise in its language.
- Figure 1: The panel d label appears to be missing.
- Line 119 and following, Subwavelength TE11 to TM11 mode conversion: The detailed mechanism for TE11 to TM11 mode conversion is not clearly explained, and it is difficult to make out precisely what is happening from the figures.
- Figure 5: I actually prefer the birdcage excitation to the true-TW excitation in some respects. The True-TW pattern still has some heterogeneity, and it appears to be less efficient in the center of the brain, where it is generally most crucial to maximize signal to noise ratio. The benefits of increased extent of excitation in the foot-head direction, meanwhile, are relatively modest, and can be outdone with appropriate transmit coil array design.
- Line 177 and onward, phase-velocity-matched SW-free TW MRI. It is unclear how robust and subject-independent this matching will be. It is also unclear how much the dielectric material (whose configuration is not shown) will have to surround the entire head to have the desired effect. Even if it is robust, phase matching only removes a problem that is generated in the first place by traveling wave excitation.
- Lines 242-244: "This breakthrough could bring us one step closer to the ideal solution of whole-body MRI at UHF which has remained as an unresolved technical challenge in the community for over three decades." It is true that whole-body UHF MRI has been both desirable and elusive, but the True-TW approach is not likely to make it much more feasible. Indeed, it would be surprising if the results shown in the head were generalizable to the entire body, where dielectric interactions with diverse tissue interfaces have already been shown (e.g. in Ref 31 of the paper's reference list) to dominate RF field distributions. And if the True-TW approach is only easy to control in the head, then it feels like overkill as a means of achieving marginally increased excitation homogeneity.
- Lines 246-7: "our findings on subwavelength mode conversion also have potentials to advance technologies in nano-scale photonics and sensors." This is mentioned twice, but never expanded upon. How might it advance such technologies?

- Lines 302-305: "The standard RF coil for clinical human head MRI at 7T, i.e., Nova 1Tx/32Rx head coil (Nova Medical, MA, US) was used for comparison, and its transceiver mode was selected for fair comparison (32-channel receive array was disabled during 305 data acquisition)." It is unclear why this markedly inferior receive configuration was selected. The receive coil would be the same for both excitations, and in fact the array sensitivities, which are brighter towards the edge of the head, are known to balance center brightening, at least when it comes overall image intensity. One other limitation of traveling-wave coils, by the way, is that they are generally quite inefficient as receivers, since they receive noise from large volumes of the body.
- Grammar and syntax could use some detailed editing throughout.

Reviewer #2 (Remarks to the Author):

In this study, the authors demonstrate a new approach to beneficially utilize traveling wave for RF field distribution in MRI applications. The prospects of the traveling wave efficient implementation in MRI can have high impact on the field. Moving to high magnetic field in MRI provides increased SNR and thus the achievable resolution in human imaging can be dramatically improved – in disease and basic science. However, one of the major challenges is the RF field inhomogeneity with the existing RF coils methods. Traveling wave can offer better control of the RF field distribution. However, the traveling wave implementations until now were not efficient enough for practical use. This study demonstrate an approach that can harness traveling wave with efficient implementation. With this approach, the traveling wave implementations can be boosted and provide a substantial step forward.

The method includes two important achievements: a) the approach offers a method to convert the TE₁₁ mode to TM₁₁ mode of the traveling wave modes, which can provide an increased efficiency for MRI due to the optimal magnetic field distribution for spins excitation. 2) a way to achieve optimal phase-velocity matching, thus adding to the increase in the efficiency of the implementation.

However, there are few major points that need clarification and deeper explanation before publication. It is not very clear why calling the existing implementation as pseudo-traveling wave and the new as true-traveling wave has merit. As far as I understand the existing traveling wave implementations in MRI are not efficient, but still these are based on well-known traveling wave modes, so it is not clear why to call it pseudo-traveling wave, while it is also not clear why to call the new implementation as "true TW". It is important to describe the properties of the mode converter better and the effect of the different parameters. It is important to report what were the reference voltages used for different scans, since this was the main well known disadvantage of the traveling wave approach until now. Additional English editing and figures details is required.

Below are the detailed comments that require addressing before publication.

Detailed/additional comments:

Abstract

It is not very clear why calling the existing implementation as pseudo-traveling wave and the new as true-traveling wave has merit. I didn't find any similar definitions from other fields. As far as I understand the existing implementation uses TE₁₁ mode, which is not optimal for MRI, as the authors mention. However, calling it pseudo-traveling wave and the new approach in which the authors demonstrate mode conversion true-traveling wave is misleading. If there is any similar definitions from other fields, please, add clear explanations to that naming.

Introduction

Line 100. The same is during the text in introduction. What is "pseudo-TW excitation"? I didn't find any relevant use of this definition in the literature and I don't see why the use of TE₁₁ results in pseudo-TW, it is just not efficient.

Line 109 "we demonstrate the feasibility of achieving true-TW MRI at a subwavelength scale" – what do you mean here by subwavelength scale? Compared to air? Please, elaborate.

Lines 121-122 "In MRI, nuclei spins are excited exclusively by circular polarized transverse magnetic

field" – this is not accurate. There is no problem to excite spins with linear polarized or other methods to generate magnetic fields. However, the optimal efficiency will be with circular polarization (at least at low fields). I suggest to rephrase.

Lines 135-145 vs lines 146-150 – what are the actual differences in the implementation of the two conversion modes that are described here. It is not clear in the text, if there are few implementations of the conversion modes, what was the option that was actually used. In Figure 1, there are three options and not two. Need to improve the description here. Or does it behave as a combination of all options.

To better understand the mode conversion capabilities, the description of the mode conversion requires better characteristics - what is the dependence of the conversion mode versus permittivity, conductivity, the diameter, the length of the dielectric?

Line 162- 163 – "was reduced by approximately 8 times inside the dielectric cylinder" – please, clarify here reduce compared to what.

Lines 167-169 "The mode converter had a bandwidth of ~ 3 MHz at the working frequency of 300 MHz, requiring precise manufacturing of the dielectric cylinder" – how the bandwidth was defined. The simulated plot looks relatively flat.

Lines 167-169 What do you mean by "precise manufacturing" – what and how affects the precision of the converter here and what required your special attention in the implementation. Need some additional information.

Line 171-173 - "the ... converter ... can also achieve targeted MRI with minimized aliasing ghosts" – what ghosts? How did you eliminated ghosts? - I assume that you mean that the B1 outside the dielectric material is low enough, so no aliasing from outside the FOV is expected. In general, this statement is problematic, since it depends on the FOV of the scan. I suggest to remove this statement. The more important statement is the increased coverage that you can get compared to existing coils based on standing waves.

Lines 188-190 – "To remove the SW effect, we introduce a high-permittivity material placed around the human head to achieve" – It sounds like you added another set of dielectric materials. I assume it is the same dielectric as in previous step. I suggest to rephrase the description here to something like – "the same dielectric material also serves here to remove the SW effect, since placing it around the human head provides the desired phase-velocity matching. "

It is important to add an analysis that compares the new approach with the traveling wave state of the art implementation in MRI. In the reference by Andreychenko, A. et al. that was mentioned by the authors (Improved steering of the RF field of traveling wave MR with a 386 multimode, coaxial waveguide. Magn. Reson. Med. 71, 1641–1649 (2014).) improved efficiency was achieved. How is the new approach compared to this method?

The authors used sucrose solution for the mode converter. The electric permittivity of the solution was mentioned, but not the conductivity in real setup. Usually such solution will have relatively high conductivity. Is there an effect of degradation in the performance or some advantage in the realistic conductivity. What is the effect of the sucrose conductivity?

What is the effect of the diameter of the dielectric on the B1 efficiency?

Since until now the main disadvantage of TW was the reference amplitude that was out of reach, especially for 180 degree pulses, it is important to write what were the reference amplitudes used for

the new approach compared to the standing wave case. Please, also show the power/transmit calibration – a plot of signal intensity vs. the square root of input power (or transmit amplitude) for Nova coil setup and the new TW implementation.

Fig3. – why it says S13, S23? In the text it says S21. Need some explanations between the demonstrated ports in figure 3b and its relevance to Fig.3 d and c and the connection to the described in the text.

There is no clear connection between the described in the text about the mode converter options and the demonstrated in Fig.1.

Method

Lines 306-309 “anthropomorphic head phantom” – is there a reference to the content of the phantom. If not, please, describe the phantom properties. What were the electrical properties of this phantom (permittivity, conductivity).

Line 308-309 “Reference transmits voltages for driving the patch antenna and the Nova coil were adjusted prior to scans.” – what were the transmit voltages for birdcage and the new method.

Lines 310-314: What was the flip angle of the GRE scan. Was the commercial GRE sequence used here? Was acceleration used? What was the scan duration, what was the FOV of the scan?

Line 314- I suggest to rephrase the sentence “... evaluate easily the excitation homogeneity” , since this scan is also T2* weighted, it mainly shows qualitatively the method coverage and excitation distribution.

The text also requires extensive English editing. Below few examples:

Line 152 “In addition to maximizing...”- you mean something like, “To maximize the transverse field, we improved...”

Line 171 “In addition to enhancing excitation efficiency” – you mean “in order to increase excitation efficiency”.

I think you should correct “cooper” to “copper”.

Most of the figures require thorough review of the figure captions and the details shown in the figure – such as the x and y axis in plots, there are several color-bars that the shown values are cut, as well as some text that is cut. Below few details:

Fig. S3 – what are the units?

Fig.1 – requires thorough review of the figure caption to properly explain what is shown. Also, where is d in the figure?

Fig.2 – required thorough review of the figure caption.

Fig.3 requires thorough review- both the caption and the plots. What is shown in c and d, what are the axes and units in plots c and d? The legend is cut

Fig.4 – what is the scale of the B1 maps shown on the right – the values are cut.

Fig. 5- there is a discrepancy in the values that are shown in fig 5a and 5b – probably due to some ranges of the phase values, for example figure a shows angles close to zero, when figure 5b shows values close to 360. Need some consistency or explanation.

Reviewer #3 (Remarks to the Author):

This paper by Gao et al. is a study towards travelling wave MRI at 7T. Travelling wave MRI has been a topic of research 10 years ago with the promise of providing an alternative method for RF excitation at high fields where standing wave effects and efficiencies of conventional RF transmit become severely detrimental.

The authors have designed a dielectric insert (a so-called subwavelength mode converter as they called it) that can convert the dominant TE₁₁ mode in the MRI bore to the TM₁₁ mode that provides higher useful transverse magnetic field. Furthermore, they claim that this insert can also reduce the standing wave (SW) effect. I assume they mean the longitudinal SW effect that occurs to counter propagating wave caused by reflections in the waveguide.

I have some reservations to the originality of the work. There has been previous work that studied the use of dielectric insert to reduce standing wave effects (Foo et al, MRM 1992) for birdcage coil and improved RF steering/homogeneity and efficiency for travelling wave MRI (Andreychenko et al, Improved RF performance...MRM 2013, Andreychenko et al, Improved steering of the RF field...., MRM 2014). I

Furthermore, I think the authors sometimes loosely formulate hypotheses that are not really substantiated by results or explained what it means (e.g. L84/85, L182, L191)

Above all, the experimental evidence is limited to some qualitative measurements (gradient echo images) of the head. I miss quantitative B₁ (amplitude and phase) measurements in the phantom and head that are in line with the electromagnetic simulations. For example, the authors claim that the TM₁₁ mode provides much higher efficiency as shown in simulations. However, they should prove this in measurements.

Finally, the authors claim that the SW are reduced. However, this only applies to longitudinal SW, while the transverse SW are most problematic. Furthermore, the residual inhomogeneity is still too large to provide useful images.

We are very grateful for the constructive review comments. We have revised the manuscript accordingly, and all revised sentences are marked with yellow color in the manuscript. Below we address each of the reviews' comments, and our responses are printed in blue.

Reviewer 1:

Overall comments: This paper returns to a decade-and-a-half-old concept of traveling wave MRI, in which the cylindrical bore of the MR scanner serves to propagate RF fields required for spin excitation and signal generation. The authors propose a mode-conversion mechanism to improve the performance of traveling wave excitation. While this mechanism is interesting (though not convincingly explained in the paper), the resulting excitation field distributions are not likely to be of much practical use in research or clinical MRI. The authors overstate the problem of inhomogeneous excitation, for which there are already some reasonable mitigation strategies. Meanwhile, the excitations their traveling-wave transmitter does produce are only marginally improved over a simple birdcage coil excitation, which can already be improved upon substantially, in the head at least, with relatively simple coil arrays and RF shimming. Moreover, the results presented here are unlikely to be generalizable beyond the head, given the interactions of RF fields with the complex dielectric structures in the rest of the body. In light of these limitations, I do not believe that the True-Travelling-Wave approach proposed in this paper will have much sway in MRI circles. Potential applications of related technologies in nano-scale photonics and sensors are mentioned, but not elaborated upon, so it is also difficult to assess any broader impact.

Authors' Response:

1. "... the resulting excitation field distributions are not likely to be of much practical use in research or clinical MRI".

- We don't agree with reviewer's comment.

We have already shown the efficacy of our proposed method through MRI experiments on both phantom and in-vivo human subjects (Fig. 8 in revised manuscript). All MRI experiments were conducted under the clinical mode (with single-channel RF transmission) of a human 7T MRI scanner (Siemens MAGNETOM), rather than using its research mode (parallel RF transmission) which needs to take rigorous yet more complex RF power calculation and management, and promising results have shown improved B_1^+ homogeneity over the canonical birdcage coil.

2. "The authors overstate the problem of inhomogeneous excitation, for which there are already some reasonable mitigation strategies...which can already be improved

upon substantially, in the head at least, with relatively simple coil arrays and RF shimming.”

- We don't agree with reviewer's comment.

Even though the RF shimming technology together with the coil array mentioned by the reviewer has shown its feasibility in mitigating B_1^+ inhomogeneity in human head imaging at 7T MRI, instead of acting as “simple coil arrays and RF shimming”, it suffered heavily from complex system architecture and operation procedure as well as major safety risks arising from complex SAR management. Due to the absence of mature SAR management method under multi-channel transmission mode, RF shimming alongside with its associated complex multi-channel transmission system have not been allowed to operate in clinical mode by the FDA. As mentioned in the “Introduction” section of our manuscript: “complex B_1^+ control methods elevate the risk of specific absorption rate (SAR), increase system complexity, and imposes negative impact over MR signal acquisition stability due to motion effects.”

In addition, current implementations of RF shimming mainly rely on resonators placed in close vicinity to the subject. Strong loading effect determines that RF calibration is necessary on a subject-specific basis considering the variations of subjects' head size, anatomy, position, etc. However, B_1^+ maps require relatively long scan time (several minutes) and it can be even longer with massive transmit coil array; as a result, subject motion will inevitably lead to dynamic changes in B_1^+ field, E field and SAR – current multi-channel transmit system are incapable of addressing the instability issues; therefore, the RF shimming technology together with resonator coil array mentioned by the reviewer may not be a suitable solution which can be widely used especially in clinical scenarios.

In contrast, employing insensitive transmission antenna driven by a single-channel transmit system is a more feasible solution in practical clinical use. Due to the nature of large field coverage and load insensitivity, the travelling wave transmission method has shown its efficacy as a whole-body transmit coil. The major limits however preventing its practical applications are: 1) low transmission efficiency and 2) lack of efficient field manipulation method for uniform excitation in complex dielectric tissues – the proposed mode-conversion method has shown great potential in addressing these two major issues in the manuscript.

Ultimately, “The problem of inhomogeneous excitation” has still been an unmet need in ultra-high field MRI. As mentioned at the very first of our manuscript: “Uniform excitation of nuclei spins through circular polarized transverse magnetic field (B_1^+) is crucial for generating favorable tissue contrast efficiently in magnetic resonance

imaging (MRI)". Uniform excitation is the basis for all MRI applications and therefore shall not be an overstated problem.

3. "... marginally improved over a simple birdcage coil excitation..."

While the birdcage coil has been the canonical design widely used as body transmit coil for mainstream MRI systems at 1.5T and 3T, it has also been serving as the standard design for head imaging in the clinical mode at 7T. As discussed above, RF shimming technology and coil arrays which require multi-channel RF transmit management are only allowed to use in the research mode.

Our study aims to devise a single-channel transmit system to be potentially used in clinical mode of future ultra-high field (UHF) ($\geq 7T$) MRI systems. Therefore, the birdcage design would be a good benchmark for comparison.

4. "Potential applications of related technologies in nano-scale photonics and sensors are mentioned, but not elaborated upon, so it is also difficult to assess any broader impact."

Subwavelength artificial structures, mode-conversion and polarization transformation have wide applications in optics and telecommunications^[1-3]. Our study has revealed a novel subwavelength dielectric material that can produce high-order mode RF magnetic field in a small aperture waveguide (below cut-off TE₁₁-to-TM₁₁ mode conversion). It has also shown capability of power-focusing, wave-impedance match as well as phase-velocity match. All features suggest it would have a broader impact beyond MRI.

References:

- [1] Luo, X. Subwavelength Artificial Structures: Opening a New Era for Engineering Optics. *Adv. Mater.* 31, 1804680 (2019).
- [2] Dorrah, A. H., Rubin, N. A., Zaidi, A., Tamagnone, M. & Capasso, F. Metasurface optics for on-demand polarization transformations along the optical path. *Nat. Photonics* 15, 287–296 (2021).
- [3] Heinrich, M. et al. Supersymmetric mode converters. *Nat Commun* 5, 3698 (2014).

Detailed comments:

- Abstract, lines 38-39, and also Introduction, lines 78-79: "Uniform B₁₊ is key to producing favorable tissue contrast..." I would not actually characterize B₁₊ uniformity as 'key,' since a wide range of transmit coils and inhomogeneity mitigation techniques exist, and since some applications do not even require uniform contrast. Uniformity is 'helpful,' perhaps, but not key.

Authors' Response:

We don't agree with reviewer's comment.

The Bloch equation [1], which was named after physicist Felix Bloch who won the Nobel Prize for his discovery of magnetic resonance phenomena, describes almost all behaviors seen in MRI:

$$\frac{d\vec{M}}{dt} = \gamma\vec{M} \times \vec{B}_{ext} + \frac{1}{T_1}(M_0 - M_z)\hat{z} - \frac{1}{T_2}\vec{M}_\perp$$

$$\vec{B}_{ext} = B_0\hat{z} + B_1\hat{x}'$$

According to the Bloch equation, it is important to ensure a uniform \vec{B}_{ext} , including both static magnetic field B_0 , and clock-wise transverse alternating magnetic field (B_1) in radio-frequency, within the region of interest. Otherwise, spatial tissue contrast information related to T1 (longitudinal relaxation) and T2 (transverse relaxation) will be contaminated by spatial information of \vec{B}_{ext} .

It is a widely recognized fact that: the inhomogeneity contrast arising from inhomogeneity B_1^+ is one of the major issues preventing successful applications in clinical scenarios [2]. Therefore, we insist that "Uniform B1+ is key to producing favorable tissue contrast..." is a correct and precise statement.

References:

- [1] Brown, R. W., Cheng, Y. C. N., Haacke, E. M., Thompson, M. R., & Venkatesan, R. Magnetic resonance imaging: physical principles and sequence design. John Wiley & Sons. (2014).
- [2] Kamil Uğurbil. Imaging at ultrahigh magnetic fields: History, challenges, and solutions. Neuroimage, 2018, 168:7-32

- Abstract, lines 39-40, and Introduction, lines 83-85: "While the standing wave...can full meet such requirements." RF excitation in most cases does not actually involve a standing wave – just a quasi-static field distribution. At high field strength and high operating frequency, standing wave effects can contribute, but field inhomogeneity can result at least as much from coil geometry and symmetry as from actual interference of forward and reflected waves. This is partly a matter of semantics, but a paper on electrodynamic effects should be precise in its language.

Authors' Response:

The expression has been modified following suggestions.

- Figure 1: The panel d label appears to be missing.

Authors' Response:

Thanks for pointing it up. The problem was resolved in the revised manuscript. Figure 1 has been revised accordingly.

- Line 119 and following, Subwavelength TE₁₁ to TM₁₁ mode conversion: The detailed mechanism for TE₁₁ to TM₁₁ mode conversation is not clearly explained, and it is difficult to make out precisely what is happening from the figures.

Authors' Response:

We have added additional illustration of the TE₁₁-to-TM₁₁ mode conversion mechanism in figures and provides detailed description in the revised manuscript.

Thanks for pointing it up. We have added additional illustration of the TE₁₁-to-TM₁₁ mode conversion mechanism in revised Figures and provided detailed descriptions in the revised manuscript.

- Figure 5: I actually prefer the birdcage excitation to the true-TW excitation in some respects. The True-TW pattern still has some heterogeneity, and it appears to be less efficient in the center of the brain, where it is generally most crucial to maximize signal to noise ratio. The benefits of increased extent of excitation in the foot-head direction, meanwhile, are relatively modest, and can be outdone with appropriate transmit coil array design.

Authors' Response:

The center-brightening effect is a typical behavior by using birdcage excitation for brain MRI at UHF^[1]. It suggests the deviation of quasi-static condition when implementing standing-wave excitation by using a cavity resonator (birdcage coil). As discussed above, uniform B_1^+ is the key in RF transmit system design. Therefore, "less efficient" in the center of the brain suggests the alleviation of standing-wave effect, which has benefit leading to uniform B_1^+ .

The "signal-to-noise ratio" mentioned by the reviewer is a metric for RF receive system, and nothing to do with the RF transmit system.

Regarding the comment: "... can be outdone with appropriate transmit coil array design", the transmit coil array is highly dependent on the multi-channel transmit system, which has not been allowed in clinical mode of current 7T MRI systems and faces a lot of challenges and risks in practical use.

References:

[1] Kamil Uğurbil. Imaging at ultrahigh magnetic fields: History, challenges, and solutions. *Neuroimage*, 2018, 168:7-32

- Line 177 and onward, phase-velocity-matched SW-free TW MRI. It is unclear how robust and subject-independent this matching will be. It is also unclear how much the dielectric material (whose configuration is not shown) will have to surround the entire head to have the desired effect. Even if it is robust, phase matching only removes a problem that is generated in the first place by traveling wave excitation.

Authors' Response:

In principle, the permittivity of the dielectric material is chosen to be the average permittivity of the human subject, which is not supposed to vary dramatically among subjects. Therefore, the proposed method is not sensitive to subject. It is also a widely recognized fact that the filling factor of travelling wave system is much smaller than resonators which is placed in close vicinity of the subject. The detailed configurations have been illustrated in Fig. 6 in the revised manuscript, along with detailed analysis of phase velocity match, in new section "Phase velocity matching" and in Fig. 5.

We don't agree with reviewer's comment of "Even if it is robust, phase matching only removes a problem that is generated in the first place by traveling wave excitation."

According to Fig. 5 & 7b in the revised paper, the major standing wave effect inside the human head along the head-feet direction has been indicated by neighboring uniform "red" and "green" phases. Such typical standing wave phase pattern was substituted with linear evolved phase by using proposed phase-velocity match. The iconic standing wave pattern appearing as alternating node and anode in B_1^+ profile was also missing. Further improvement can be made through extending the phase-velocity match material to the neck and body part.

- Lines 242-244: "This breakthrough could bring us one step closer to the ideal solution of whole-body MRI at UHF which has remained as an unresolved technical challenge

in the community for over three decades.” It is true that whole-body UHF MRI has been both desirable and elusive, but the True-TW approach is not likely to make it much more feasible. Indeed, it would be surprising if the results shown in the head were generalizable to the entire body, where dielectric interactions with diverse tissue interfaces have already been shown (e.g. in Ref 31 of the paper’s reference list) to dominate RF field distributions. And if the True-TW approach is only easy to control in the head, then it feels like overkill as a means of achieving marginally increased excitation homogeneity.

Authors’ Response:

We don’t agree with reviewer’s comment of “the True-TW approach is not likely to make it much more feasible”.

Through using proposed subwavelength dielectric material, we have demonstrated the feasibility of enhanced travelling-wave excitation over the human head at 7T. It has shown promising results with enhanced efficiency as well as alleviated standing wave effect. It suggests the potential of artificially structured materials in electromagnetic field manipulation of travelling wave, which has rarely been investigated in travelling-wave MRI. Since the efficacy of artificially structured materials in electromagnetic wave manipulation has been well recognized, it is reasonable to extend our method to the larger human body part.

- Lines 246-7: “our findings on subwavelength mode conversion also have potentials to advance technologies in nano-scale photonics and sensors.” This is mentioned twice, but never expanded upon. How might it advance such technologies?

Authors’ Response:

As mentioned above, subwavelength artificial structures and mode-conversion have wide applications in optics and telecommunications. Our study has revealed a novel subwavelength artificial dielectric material that can produce high-order mode magnetic field in a small aperture waveguide (below cut-off TE₁₁-to-TM₁₁ mode conversion). It has also shown capability of power-focusing, wave-impedance match as well as phase-velocity match. All features suggest it may have a broader impact over MRI at UHF.

- Lines 302-305: “The standard RF coil for clinical human head MRI at 7T, i.e., Nova 1Tx/32Rx head coil (Nova Medical, MA, US) was used for comparison, and its transceiver mode was selected for fair comparison (32-channel receive array was

disabled during 305 data acquisition).” It is unclear why this markedly inferior receive configuration was selected. The receive coil would be the same for both excitations, and in fact the array sensitivities, which are brighter towards the edge of the head, are known to balance center brightening, at least when it comes overall image intensity. One other limitation of traveling-wave coils, by the way, is that they are generally quite inefficient as receivers, since they receive noise from large volumes of the body.

Authors' Response:

“It is unclear why this markedly inferior receive configuration was selected.”

Because the travelling wave system was driven in transceiver mode, for a fair comparison, we also used the transceiver mode of the birdcage coil. Transceiver mode has the advantage in conveniently qualifying the RF transmit field distribution through acquiring proton-density weighted images. According to the principle of reciprocity^[1], the transmit field profile (B_1^+) is the conjugate pair of receive sensitivity profile (B_1^-).

The signal equation \mathbf{S} of proton-density weighted image is: $\mathbf{S} = \mathbf{M}_0 \cdot \mathbf{PD} \cdot |B_1^+| \cdot |B_1^-|$, where \mathbf{M}_0 is the magnetization vector, \mathbf{PD} is the proton density.

In practical applications, the travelling wave operates in transmit-only mode while separate coil arrays are used for RF reception – this is the standard RF system architecture in mainstream clinical MRI system, and the only difference is using birdcage coil for whole-body transmit for the latter.

References:

[1] Hoult, D. I. The principle of reciprocity in signal strength calculations - A mathematical guide. Concepts in Magnetic Resonance 12, 173–187 (2000).

- Grammar and syntax could use some detailed editing throughout.

Authors' Response:

The grammar and syntax have been thoroughly improved in throughout the revised manuscript.

Reviewer 2:

In this study, the authors demonstrate a new approach to beneficially utilize traveling wave for RF field distribution in MRI applications. The prospects of the traveling wave efficient implementation in MRI can have high impact on the field. Moving to high magnetic field in MRI provides increased SNR and thus the achievable resolution in human imaging can be dramatically improved – in disease and basic science. However, one of the major challenges is the RF field inhomogeneity with the existing RF coils methods. Traveling wave can offer better control of the RF field distribution. However, the traveling wave implementations until now were not efficient enough for practical use. This study demonstrate an approach that can harness traveling wave with efficient implementation. With this approach, the traveling wave implementations can be boosted and provide a substantial step forward.

The method includes two important achievements: a) the approach offers a method to convert the TE₁₁ mode to TM₁₁ mode of the traveling wave modes, which can provide an increased efficiency for MRI due to the optimal magnetic field distribution for spins excitation. 2) a way to achieve optimal phase-velocity matching, thus adding to the increase in the efficiency of the implementation.

However, there are few major points that need clarification and deeper explanation before publication. It is not very clear why calling the existing implementation as pseudo-traveling wave and the new as true-traveling wave has merit. As far as I understand the existing traveling wave implementations in MRI are not efficient, but still these are based on well-known traveling wave modes, so it is not clear why to call it pseudo-traveling wave, while it is also not clear why to call the new implementation as “true TW”. It is important to describe the properties of the mode converter better and the effect of the different parameters. It is important to report what were the reference voltages used for different scans, since this was the main well known disadvantage of the traveling wave approach until now. Additional English editing and figures details is required.

Authors' Response:

1. “It is not very clear why calling the existing implementation as pseudo-traveling wave and the new as true-traveling wave has merit.”

To avoid misunderstanding, the term “true-traveling wave” has been replaced with “modified travelling-wave method” for “efficient travelling-wave MRI” in the revised manuscript. Our proposed “subwavelength dielectric waveguide” (re-termed in the revised manuscript) has merits in TE₁₁-to-TM₁₁ mode conversion below the cut-off limit, power focusing, wave impedance matching, as well as phase velocity matching. In conclusion, it leads to efficient implementations of TW MRI with high uniformity.

Detailed illustrations of the modified travelling-wave MRI based on subwavelength dielectric materials were given in the revised manuscript.

2. “It is important to describe the properties of the mode converter better and the effect of the different parameters.”

Details about the properties of the mode converter as well as the effect of different parameters have been added in the revised manuscript (section “TE₁₁-to-TM₁₁ mode conversion in TW MRI”; Figs 1 & 2).

3. “It is important to report what were the reference voltages used for different scans, since this was the main well known disadvantage of the traveling wave approach until now.”

Reference voltages for different scans have been given in the revised manuscript.

4. “Additional English editing and figures details is required.”

The grammar and syntax have been thoroughly improved in throughout the revised manuscript.

Below are the detailed comments that require addressing before publication.

Detailed/additional comments:

Abstract

It is not very clear why calling the existing implementation as pseudo-traveling wave and the new as true-traveling wave has merit. I didn't find any similar definitions from other fields. As far as I understand the existing implementation uses TE₁₁ mode, which is not optimal for MRI, as the authors mention. However, calling it pseudo-traveling wave and the new approach in which the authors demonstrate mode conversation true-traveling wave is misleading. If there is any similar definitions from other fields, please, add clear explanations to that naming.

Introduction

Line 100. The same is during the text in introduction. What is “pseudo-TW excitation”? I didn't find any relevant use of this definition in the literature and I don't see why the use of TE₁₁ results in pseudo-TW, it is just not efficient.

Authors' Response:

Thanks for the suggestion. To avoid misunderstanding, the term “true-traveling wave”

has been replaced with “modified travelling-wave method” for “efficient travelling-wave MRI” in the revised manuscript.

Line 109 “we demonstrate the feasibility of achieving true-TW MRI at a subwavelength scale” – what do you mean here by subwavelength scale? Compared to air? Please, elaborate.

Authors' Response:

The “dielectric waveguide” was structured in its electrical length smaller than half-wavelength; otherwise, it will suffer from SW effect due to unmatched phase velocity between itself and the air, as shown in Fig. 1 in the revised manuscript.

Lines 121-122 “In MRI, nuclei spins are excited exclusively by circular polarized transverse magnetic field” – this is not accurate. There is no problem to excite spins with linear polarized or other methods to generate magnetic fields. However, the optimal efficiency will be with circular polarization (at least at low fields). I suggest to rephrase.

Authors' Response:

The definition of effective transmit field in MRI is $B_1^+ = \mu \frac{H_x + iH_y}{2}$ which it is the clock-wise circular-polarized transverse magnetic field^[1]. Even though linear polarized magnetic field can excite nuclei spins, only clock-wise circular-polarized transverse magnetic field component works, and that is the reason why circular polarization has the optimal efficiency. Therefore, the expression “In MRI, nuclei spins are excited exclusively by circular polarized transverse magnetic field” is correct and precise.

References:

[1] Hoult, D. I. The principle of reciprocity in signal strength calculations - A mathematical guide. *Concepts in Magnetic Resonance* 12, 173–187 (2000).

Lines 135-145 vs lines 146-150 – what are the actual differences in the implementation of the two conversion modes that are described here. It is not clear in the text, if there are few implementations of the conversion modes, what was the option that was actually used. In Figure 1, there are three options and not two. Need to improve the description here. Or does it behave as a combination of all options.

Authors' Response:

Thanks for the suggestion. The description of these two paragraphs have been revised to avoid misunderstanding.

To better understand the mode conversion capabilities, the description of the mode conversion requires better characteristics - what is the dependence of the conversion mode versus permittivity, conductivity, the diameter, the length of the dielectric?

Authors' Response:

Thanks for the suggestion. Detailed analysis and description of the mode conversion have been added in the revised manuscript as instructed.

Line 162- 163 – “was reduced by approximately 8 times inside the dielectric cylinder” – please, clarify here reduce compared to what.

Authors' Response:

Thanks for the suggestion. The description has been revised.

Lines 167-169 “The mode converter had a bandwidth of ~3 MHz at the working frequency of 300 MHz, requiring precise manufacturing of the dielectric cylinder” – how the bandwidth was defined. The simulated plot looks relatively flat.

Authors' Response:

Thanks for pointing it up. This description is incorrect and has been deleted in the revised manuscript.

Lines 167-169 What do you mean by “precise manufacturing” – what and how affects the precision of the converter here and what required your special attention in the implementation. Need some additional information.

Authors' Response:

Thanks for pointing it up. This description is incorrect and has been deleted in the revised manuscript.

Line 171-173 - “the ... converter ... can also achieve targeted MRI with minimized aliasing ghosts” – what ghosts? How did you eliminated ghosts? - I assume that you mean that the B1 outside the dielectric material is low enough, so no aliasing from outside the FOV is expected. In general, this statement is problematic, since it depends on the FOV of the scan. I suggest to remove this statement. The more important statement is the increased coverage that you can get compared to existing coils based on standing waves.

Authors' Response:

Due to the large signal coverage of classic travelling wave MRI, the FOV has to be set large enough to cover all possible areas with spins excited, in order to avoid wrap-around aliasing will appear in the image along the phase-encoding direction.

Enhancing B_1^+ in the target imaging area with artificially structured materials can possibly achieve zoomed imaging. A similar idea has been reported using ceramic resonators in breast imaging ^[1] – more elaborations about the benefit have been added in the revised manuscript.

References:

[1] Shchelokova, A. et al. Ceramic resonators for targeted clinical magnetic resonance imaging of the breast. *Nat Commun* 11, 3840 (2020).

Lines 188-190 – “To remove the SW effect, we introduce a high-permittivity material placed around the human head to achieve” – It sounds like you added another set of dielectric materials. I assume it is the same dielectric as in previous step. I suggest to rephrase the description here to something like – “the same dielectric material also serves here to remove the SW effect, since placing it around the human head provides the desired phase-velocity matching. “

Authors' Response:

Thanks for the suggestion. The description has been rephrased accordingly in the revised manuscript.

It is important to add an analysis that compares the new approach with the traveling wave state of the art implementation in MRI. In the reference by Andreychenko, A. et al. that was mentioned by the authors (Improved steering of the RF field of traveling wave MR with a 386 multimode, coaxial waveguide. *Magn. Reson. Med.* 71, 1641–

1649 (2014).) improved efficiency was achieved. How is the new approach compared to this method?

Authors' Response:

Following the suggestion, we have compared our approach with the-state-of-the-art implementation of the travelling wave technique in MRI through numerical simulations, as shown in Supplement Fig. 2. Detailed discussion and analysis has been given in the revised manuscript, in sections “Introduction” and “Wave impedance matching”.

The authors used sucrose solution for the mode converter. The electric permittivity of the solution was mentioned, but not the conductivity in real setup. Usually such solution will have relatively high conductivity. Is there an effect of degradation in the performance or some advantage in the realistic conductivity. What is the effect of the sucrose conductivity?

Authors' Response:

The sucrose solution was a mixture of distilled water and sucrose with no additional ingredient. Sucrose was dissolved as molecules rather than ions, therefore the solution should ideally be non-conductive. We measured the relative permittivity and conductivity of sucrose solutions with different recipes as shown below:

Mass ratio (water/sucrose)	100:102	100:74	100:54
relative permittivity	61.7018	66.9494	70.5636
conductivity (S/m)	0.0708	0.0467	0.0241

All conductivities are below 0.1 S/m, suggesting minor electric loss which is favorable to our approach.

What is the effect of the diameter of the dielectric on the B1 efficiency?

Authors' Response:

Thanks for the question. Detailed analysis and description of the subwavelength dielectric materials including the effect of the diameter have been added in the revised manuscript as in Figs 1-5/sections “TE11-to-TM11 mode conversion in TW MRI”, “ Power focusing”, “Wave impedance matching”, and “Phase velocity matching”.

Since until now the main disadvantage of TW was the reference amplitude that was out of reach, especially for 180 degree pulses, it is important to write what were the reference amplitudes used for the new approach compared to the standing wave case. Please, also show the power/transmit calibration – a plot of signal intensity vs. the square root of input power (or transmit amplitude) for Nova coil setup and the new TW implementation.

Authors' Response:

Thanks for the suggestion. The reference amplitudes as well as power/transmit calibration results for both approaches have been provided in the revised manuscript as in Fig. 8/section “ Method/MRI experiment”.

Due to the intrinsic lower filling factor in radiative transmission compared to reactive coupling, a larger proportion of power was dissipated in feed antenna for TW excitation method in practice compared to the canonical resonator excitation method. However, it should be noted that higher filling factor leads to higher SAR due to more power dissipated in load as well. In addition, it is the SAR normalized efficiency matters for transmission. The lower filling factor as well as low-SAR characteristic of TW method makes it impractical to evaluate SAR through MRI thermometry. Therefore, the B_1^+ mapping results were normalized to 1 W power dissipated in the load, through Q-ratio ($Q_{\text{unload}}/ Q_{\text{load}}$) measurement.

Fig3. – why it says S13, S23? In the text it says S21. Need some explanations between the demonstrated ports in figure 3b and its relevance to Fig.3 d and c and the connection to the described in the text.

Authors' Response:

Thanks for pointing it up. The use of S21 in the text was incorrect. Corrected and detailed descriptions Figure 3b have been given in the revised manuscript.

There is no clear connection between the described in the text about the mode converter options and the demonstrated in Fig.1.

Authors' Response:

As discussed above, there is only one option in the implementation of the mode converter. Detailed description about the principle of mode conversion have been added

in the revised manuscript as in Figs .1-2/section “ TE11-to-TM11 mode conversion in TW MRI”.

Method

Lines 306-309 “anthropomorphic head phantom” – is there a reference to the content of the phantom. If not, please, describe the phantom properties. What were the electrical properties of this phantom (permittivity, conductivity).

Authors' Response:

Thanks for pointing it up. The reference of the phantom has been provided in the revised manuscript. The relative permittivity of the phantom is close to water which is 78, and the conductivity is 0.6 S/m.

Line 308-309 “Reference transmits voltages for driving the patch antenna and the Nova coil were adjusted prior to scans.” – what were the transmit voltages for birdcage and the new method.

Authors' Response:

Both references transmit voltages for the birdcage coil and the patch antenna are given in the revised manuscript as suggested.

Lines 310-314: What was the flip angle of the GRE scan. Was the commercial GRE sequence used here? Was acceleration used? What was the scan duration, what was the FOV of the scan?

Authors' Response:

Details about the GRE sequence scan parameters including flip angle, sequence type, acceleration, scan duration as well as the FOV are given in the revised manuscript as suggested. No parallel acceleration was used since both coils were operated under single-channel receive mode.

Line 314- I suggest to rephrase the sentence “... evaluate easily the excitation homogeneity” , since this scan is also T2* weighted, it mainly shows qualitatively the method coverage and excitation distribution.

Authors' Response:

Thanks for the suggestion. The sentence has been rephrased as “GRE T2* images ... were acquired to qualitative evaluate the excitation homogeneity.”.

The text also requires extensive English editing. Below few examples:

Line 152 “In addition to maximizing...”- you mean something like, “To maximize the transverse field, we improved...”

Line 171 “In addition to enhancing excitation efficiency” – you mean “in order to increase excitation efficiency”.

Authors' Response:

The descriptions have been adjusted accordingly in the revised manuscript.

I think you should correct “cooper” to “copper”.

Authors' Response:

The typo has been adjusted in the revised manuscript.

Most of the figures require thorough review of the figure captions and the details shown in the figure – such as the x and y axis in plots, there are several color-bars that the shown values are cut, as well as some text that is cut. Below few details:

Fig. S3 – what are the units?

Fig.1 – requires thorough review of the figure caption to properly explain what is shown. Also, where is d in the figure?

Fig.2 – required thorough review of the figure caption.

Fig.3 requires thorough review- both the caption and the plots. What is shown in c and d, what are the axes and units in plots c and d? The legend is cut

Fig.4 – what is the scale of the B1 maps shown on the right – the values are cut.

Fig. 5- there is a discrepancy in the values that are shown in fig 5a and 5b – probably due to some ranges of the phase values, for example figure a shows angles close to zero, when figure 5b shows values close to 360. Need some consistency or explanation.

Authors' Response:

Following above suggestions, all figures have been thoroughly revised with problematic descriptions corrected.

Reviewer 3:

This paper by Gao et al. is a study towards travelling wave MRI at 7T. Travelling wave MRI has been a topic of research 10 years ago with the promise of providing an alternative method for RF excitation at high fields where standing wave effects and efficiencies of conventional RF transmit become severely detrimental.

The authors have designed a dielectric insert (a so-called subwavelength mode convertor as they called it) that can convert the dominant TE₁₁ mode in the MRI bore to the TM₁₁ mode that provides higher useful transverse magnetic field. Furthermore, they claim that this insert can also reduce the standing wave (SW) effect. I assume they mean the longitudinal SW effect that occurs to counter propagating wave caused by reflections in the waveguide.

I have some reservations to the originality of the work. There has been previous work that studied the use of dielectric insert to reduce standing wave effects (Foo et al, MRM 1992) for birdcage coil and improved RF steering/homogeneity and efficiency for travelling wave MRI (Andreychenko et al, Improved RF performance...MRM 2013, Andreychenko et al, Improved steering of the RF field...., MRM 2014).

Authors' Response:

We agree with the fact that dielectric materials and artificial materials are well recognized in manipulating electromagnetic field distribution and wave propagation. However, methodologies in references mentioned by the reviewer are fundamentally different from our approach. The dielectric waveguide structure has been firstly introduced in MRI to enhance wave propagation in electrically large human bodies, and has shown breakthrough effects in improving TW MRI excitation efficiency and homogeneity.

1. (Foo et al, MRM 1992) This study focused on manipulating electromagnetic field distribution at reactive near-field regions, in which dielectric materials were added in the coil-to-shield space of a birdcage coil (a type of cavity resonator). However, the working principle of dielectric materials on manipulating electromagnetic fields are different at reactive near-field (quasi-static) regions and radiative regions. The former is based on capacitively coupling as well as faraday induction, where electric field and magnetic field are disassociated; for the latter case, electric and magnetic fields are highly associated.

Therefore, the idea proposed in this study is fundamentally different with ours which attempted to address standing wave issues at radiative regions. To the best of our knowledge, our study firstly demonstrated the feasibility of standing wave

reduction in travelling MRI by using dielectric inserts.

We have therefore compared its performance with our approach through numerical simulation, and the results have been provided in the supplementary part of the revised manuscript.

2. (Andreychenko et al, Improved RF performance...MRM 2013) This study presented an efficient implementation of travelling wave MRI through adding conductive cylinder inset for wave impedance match. However, the giant conductive cylinder is inconvenient to handle and heavily suffers from eddy currents due to fast switching gradients polarities during scans. In addition, the mechanism of standing wave effect was not revealed and the issue was not well addressed.

We have therefore compared its performance with our approach through numerical simulation, and the results have been provided in the supplementary part of the revised manuscript.

3. (Andreychenko et al, Improved steering of the RF field....., MRM 2014) This study presented a method of propagating multiple higher-order mode magnetic field through adding dielectric tubes surrounding imaging subject. It was reported that multiple modes propagate simultaneous, which is however adverse to the field homogeneity, and thus multi-channel RF shimming had to be employed to achieve uniform B_1^+ . Such method is not as efficient as our approach in which single optimal mode (TM_{11}) was converted and used. To be specific, in our study, uniform B_1^+ field can be efficiently produced by using the clinical mode (single-channel RF transmission), instead of using complex multi-channel RF transmission (the research mode).

In summary, although above studies (Andreychenko et al, Improved RF performance...MRM 2013, Andreychenko et al, Improved steering of the RF field....., MRM 2014) have improved the travelling wave MRI in the aspect of efficiency, but they still suffered from complex hardware configuration or eddy current effects, and the importance of transverse magnetic mode (TM_{11}) in efficient and uniform excitation was not raised yet.

Our study aimed to address the problem of TE_{11} -to- TM_{11} mode conversion below the cut-off limit in a waveguide naturally embedded within a human MRI, which is fundamentally important to propagate single transverse magnetic mode with maximized efficiency in producing B_1^+ with high uniformity. Compared to state-of-the-art implementations of TW MRI, our method can achieve effective TE_{11} -to- TM_{11} mode conversion, power focusing, wave impedance matching, as well as phase velocity

matching simultaneously but with single subwavelength dielectric waveguide structure only.

Furthermore, I think the authors sometimes loosely formulate hypotheses that are not really substantiated by results or explained what it means (e.g. L84/85, L182, L191)

Authors' Response:

Following the suggestion, detailed elaborations have been provided in the corresponding context.

Above all, the experimental evidence is limited to some qualitative measurements (gradient echo images) of the head. I miss quantitative B_1 (amplitude and phase) measurements in the phantom and head that are in line with the electromagnetic simulations. For example, the authors claim that the TM_{11} mode provides much higher efficiency as shown in simulations. However, they should prove this in measurements.

Authors' Response:

The quantitative B_1 maps based on AFI sequence were acquired in the phantom by using both birdcage coil and the modified TW method, and the results have been provided in the results figures. Since only magnitude of B_1^+ determines the flip angle of spin excitations, and complex RF shimming was not utilized in this study, we only provided quantitative B_1 results in its amplitude.

Finally, the authors claim that the SW are reduced. However, this only applies to longitudinal SW, while the transverse SW are most problematic. Furthermore, the residual inhomogeneity is still too large to provide useful images.

Authors' Response:

According to phase maps shown in Fig. 7b (revised manuscript), the standing wave effect in the human head mainly exists in the longitudinal direction, as indicated by the line of the fast 180° phase change (red-green boundary) and its neighboring uniform phase distribution. Through phase-velocity matched TW, we could nonetheless achieve uniform B_1^+ in both longitudinal and transverse directions, as shown in Fig. 7b and Fig. 8. The typical center-brightening effect can be effectively reduced.

We do not agree with that “the residual inhomogeneity is still too large to provide useful

images.” We have demonstrated higher uniformity of B_1^+ compared to the canonical birdcage coil through both of numerical simulations and in-vivo MRI experiments (quantitative B1 maps and qualitative anatomical images as show in Fig. 8). Considering the canonical birdcage coil has been widely used worldwide for head imaging at 7T (in the clinical mode), our approach can definitely provide “useful images” as well.

Reviewer #1 (Remarks to the Author):

The authors have put a great deal of effort into this revision of a manuscript which did not originally convince me of its impact. Their new explanations of the physical phenomena and design principles underlying their modified traveling wave system are much clearer now and, as a result, much more interesting. The extensive additional text and figures provided – assuming they do not exceed the journal's length limits – should pique the interest of those who, like me, are aficionados of electrodynamics in MR. The experimental results, however, are somewhat less exciting. Only modest improvements over standard birdcage performance are shown. Moreover, while the authors explore the dependence of transmit performance on various parameters of their dielectric waveguide, they compare against only a single birdcage design, raising the question of whether the observed gains would disappear when comparing to an optimized birdcage design.

My other note, which I will frame as advice to the authors, is that they have not made it easy for reviewers like me to appreciate the work they have put in. Their written response to many of my original concerns is simply "we disagree," followed by reference to very general principles or rehashing of the Bloch equations. The authors are, of course, entitled to disagree with anything a reviewer says. On the other hand, a reviewer (especially one well versed in the Bloch equations not to mention the practical realities of high-field MR) is also entitled to the opinion that the authors have not taken their response seriously. I confess that the rebuttal document put me off enough that I almost missed the substantive changes to the manuscript (e.g., its new focus on efficiency as well as homogeneity, its extensive new field plots and explanations of mode conversion, etc.). Meanwhile, various features of the revised manuscript make it more difficult to review than it needs to be. The changed sections are all highlighted uniformly in yellow, making much of the new manuscript a sea of yellow. At least some marginal references to particular reviewer comments would have been helpful. Also, the figures, while informative, are quite busy, and more guidance in the captions about key features to note, or even about the meaning of various symbols, would also have been welcome. Figures are also cited out of order in several places. Grammar and syntax, especially in the revised sections of text, continue to require some editing.

Overall, I am more enthusiastic about the revised manuscript than I was about the original. I am still not convinced that traveling wave systems with subwavelength dielectric waveguides will rescue UHF MRI from challenges of B1+ homogeneity or efficiency – or even that UHF MRI needs as much rescuing as the authors claim it does. It is still not clear to me that modified traveling wave systems are more practical than simple birdcage transmit coils or, say, multicoil transmit arrays with fixed shims, which also require only a single transmit channel. However, I do find the principles of operation of the guided traveling wave system presented here to be thought-provoking, and I expect I will not be alone in this.

Reviewer #2 (Remarks to the Author):

This study demonstrates an approach that can harness traveling wave with efficient implementation, which therefore provides important insights for the field development. The authors improved significantly the article with better description and characterization of the method. However, there are still two main concerns that needs addressing: 1) over the text the authors mention several times that the TW with single-channel transmission can be used as a solution for uniform B1 distribution at ultra-high field MRI. I don't think that this was demonstrated in previous works (as stated) and it was not demonstrated in this work. The authors achieved in this work similar RF efficiency and better homogeneity by 20% compared to a birdcage coil. Many other works also demonstrated that both efficiency and homogeneity can be improved with different RF coil designs. Although what authors showed here is an important achievement, it does not show that uniform distribution can be achieved with single channel transmission. This should be described more carefully in the text.

2) Another concern that there are still many details/descriptions in Figures and in the text that require correction.

Below the detailed/specific comments:

Abstract

The abstract is somewhat long, repeating similar statements of "uniform spatial disturbing that is a key...", then repeating it "Uniform excitation under single-channel-transmission mode has been an unmet need ", while the results or conclusions does not offer a solution. There is an improvement of 20% of B1 inhomogeneity. So, I suggest to remove some parts of the repeated statements.

"For the wave impedance match capability of the dielectric waveguide guide, opposed to unmatched 20% power transfer efficiency, up to 70% stimulated power was delivered to the load under well-matched condition" – this is not very clear sentence.

Introduction

p.3 l.103 – "the advantage of TW excitation in producing uniform B1 has been discovered ..." – here the sentence is based on an article by Foo, however the article discusses RF field homogeneity in frequencies of 64 and 170 MHz. I am not sure at all that the homogeneity that was described is relevant for 297MHz. However, there are more recent works that showed that TW can be used to manipulate the distribution (targeting uniform distribution) by combined several TW modes (which is similar to multi-channel transmission with the SW approach).

p.3 l.110-118 – It would be useful to add here in the introduction two references that used similar to the current work approach (although without detailed analysis) - in one passive coils were used to improve local RF field distribution when combined with TW transmission (1) and in another a dielectric was added to increase the RF field in a target region (2).

1. X. Yan, X. Zhang, J C. Gore, and W. A. Grissom . Improved traveling wave efficiency in 7T human MRI using wireless local loop and dipole arrays. Proc. Intl. Soc. Mag. Reson. Med. 25 (2017), 4291.

2. R. Schmidt, A. Webb. Improving travelling wave efficiency at 7 T using dielectric material placed "beyond" the region of interest. Proc. Intl. Soc. Mag. Reson. Med. 24 (2016) . 3532

p.4.126 "In MRI, nuclei spins are excited exclusively by circular polarized transverse magnetic field" – I still think that the sentence is misleading. So I suggest rephrasing it to something like "In MRI, nuclei spins are excited exclusively by the effective circular polarized transverse magnetic field" or In MRI, nuclei spins are excited exclusively by circular polarized component of the transverse magnetic field".

Fig.2 The images in d show some permittivity values, while the plot in b shows different set of values. It is somewhat confusing. Was there some reason to show different values? The plots in c show dependence in wall thickness. It would be useful to mention that it is shown in Fig.1d (as far as I understand). And what was the permittivity in c plot. What was D in plot b and for images in d.

Fig.3 no distances are shown in a and b), so it is hard to appreciate where are the relevant cross sections that are shown in c.

p.6 l.169 – "distilled water tubes with high dielectric constant " – I suggest to make it more clear that this was done in another study, not in the current study, otherwise it is not clear here.

p.8.l.199 "Here, we demonstrate the dielectric waveguide structure can also be used for wave impedance matching" – "we demonstrate that"

p.8.l.204 "is lowered through increasing the length"- needs English editing

p.9.l.215 – "incoherent wave front ...triggers ... and leads to the formation SW" – maybe "formation of SW" or "formation of the SW". It requires English editing.

Fig.6 – "multiple dielectric cubes" – did you use different numbers in different cases. If so, please, write how many cubes were used for each case or write here instead of multiple – how many cubes were used here.

In the image it says that the permittivity was 21 and 52 and in the text 21 and 55. Fig.5 shows results for permittivity of 25 and 55 . Is it also a mistake or why different figures show different values. I would suggest that for consistency, if the optimal solution was based on permittivity of 21 to show that case in all figures.

In the figure caption it says S31,S32 and in the image it says S13, and so on.

p.10.l.238 "As discussed above, TW excitation is a promising solution to produce uniform B1". It was not really discussed above. As far as I understand to achieve uniform (or close to it) distribution one needs to combine several modes as was demonstrated in previous works with TW, however this is somewhat equivalent and similar in its complexity to the multi-channel transmit methodology that is currently developed.

p.11 l.250 "multiple dielectric cubes" – please, write here how many cubes were placed.

p.12 l.283 – 287 "It should be noted ..." – very long sentence that is hard to follow. I would suggest to rephrase.

Methods

p.14 l.348-350 – First sentence here describes the dielectric waveguide array with permittivity of 52 and the next sentence say that "all water cubes". Water has permittivity of 78. I assume it was not water cubes, but a suspension of water and sucrose as was mentioned in other parts of the texts. So "water cubes" is misleading.

In "Subwavelength dielectric waveguide prototype" section:

First paragraph says that the cubes were filled with water. But it also says that sucrose was used to lower its permittivity. But it does not say to lower to what and what was the different materials percents in the mixture, as well as what were its measured final permittivity and conductivity. Then the next paragraph is highly confusing, since it repeats about the containers – are these different containers? But then it does not mention sucrose anymore.

Reviewer #3 (Remarks to the Author):

I would like to thank the authors for their thorough review. However, although their results showed a slight increase in efficiency and homogeneity over previous travelling wave approach with CP excitation, the increase in performance is marginal in my belief.

The authors take a CP birdcage or CP mode as benchmark but this is not the state-of-the art. Over the last decades multi-transmit 7T head coil technology has evolved and nowadays a 8 channel head coil with RF shimming is a standard procedure and provides superior B1+ control than shown here. Many of the workflow related issues have been largely solved for multi-channel 7T head imaging. It does not warrant a large investment into travelling mode direction anymore...

Happy New Year!

We thank the reviewers for their careful assessments of this study and their insightful and helpful comments. We have substantially revised the manuscript according to the reviewer's comments. All revised sentences are printed in **Red** color in the manuscript. Below we address each of the review comments, and our responses are printed in **Blue**.

REVIEWER #1

Comment #1

The authors have put a great deal of effort into this revision of a manuscript which did not originally convince me of its impact. Their new explanations of the physical phenomena and design principles underlying their modified traveling wave system are much clearer now and, as a result, much more interesting. The extensive additional text and figures provided – assuming they do not exceed the journal's length limits – should pique the interest of those who, like me, are aficionados of electrodynamics in MR.

Authors' Response:

We are very grateful to the constructive comments and valuable advice in writing.

Comment #2

The experimental results, however, are somewhat less exciting. Only modest improvements over standard birdcage performance are shown. Moreover, while the authors explore the dependence of transmit performance on various parameters of their dielectric waveguide, they compare against only a single birdcage design, raising the question of whether the observed gains would disappear when comparing to an optimized birdcage design.

Authors' Response:

Since the wavelength (12 cm - 20 cm at 297MHz) in human body is very close to the dimension of human head (~15 cm), B1+ magnitude variation caused by standing-wave (SW) effect becomes prominent, and the canonical birdcage coil designed at quasi-static region produces center-brightening phenomenon inevitably – such effect has been widely observed in 7T MRI images acquired by conventionally equipped birdcage head coil (Noval Medical) in clinical scenarios [1-3]. It is intrinsically determined by the standing wave produced by its volume resonator design, whereas to the best of our knowledge, few studies have shown that such effect can be effectively inhibited through optimizing the birdcage design. Specifically, the underlying mechanism of travelling wave (TW) in achieving uniform B1+ distribution is fundamentally different with volume resonators.

For a normalized traveling wave f_{TW} with angular frequency ω , it can be explicitly expressed as the function of spatial coordinate \mathbf{r} and time t as follows:

$$f_{TW}(\mathbf{r}, t) = \sin(\omega \cdot t + \mathbf{k} \cdot \mathbf{r}) \quad (1)$$

where $k=2\pi/\lambda$ is the wave number along propagation direction, λ is the wavelength. A standing wave f_{SW} can be expressed as the superposition of two TWs propagating in opposite directions as follows:

$$f_{SW}(\mathbf{r}, t) = f_{TW}^+ + f_{TW}^- = \sin(\omega \cdot t) \cdot \cos(\mathbf{k} \cdot \mathbf{r}) \quad (2)$$

According to equations (1) and (2), the magnitude of travelling wave is uniform in space, while the magnitude of standing wave varies over space as the function of $\cos(\mathbf{k} \cdot \mathbf{r})$.

Since the magnitude variation “ $\cos(\mathbf{k} \cdot \mathbf{r})$ ” arises from the nature of standing wave, inhomogeneous B1+ magnitude produced by birdcage resonators cannot be simply alleviated through optimization. In contrast, by using the proposed dielectric waveguide TW method, although with modest improvements indicated by 20% reduction in the RMSE of B1+ magnitude in the human head, as shown in revised Fig.9 f,h (original Fig.8), the well-known center-brightening phenomenon has been effectively inhibited.

Discussions about intrinsic differences between standing wave vs. travelling wave in B1+ magnitude variation in space has been added in the “Introduction” section. A new figure was drawn and has been added as Fig.1 to better describe and compare the RF excitation systems at 3T vs. 7T, and via standing wave (volume resonator) vs. travelling wave, as shown below.

[Introduction]

Compared to the SW, the travelling wave (TW) is naturally uniform in magnitude in space (shown in Fig. 1). For a normalized traveling wave f_{TW} with angular frequency ω , it can be explicitly expressed as the function of spatial coordinate \mathbf{r} and time t : $f_{TW}(\mathbf{r}, t) = \sin(\omega \cdot t + \mathbf{k} \cdot \mathbf{r})$, while a standing wave f_{SW} can be expressed as the superposition of two TWs propagating in opposite directions: $f_{SW}(\mathbf{r}, t) = f_{TW}^+ + f_{TW}^- = \sin(\omega \cdot t) \cdot \cos(\mathbf{k} \cdot \mathbf{r})$. It is clearly to see that the magnitude of TW is a constant. In contrast, the magnitude of SW varies in space as the function of $\cos(\mathbf{k} \cdot \mathbf{r})$, where $k=2\pi/\lambda$ is the wave number along propagation direction, λ is the wavelength. B1+ inhomogeneity residing in SW scales up with increased working frequency.

Fig.1. Schematic diagrams of RF transmission systems at 3T and 7T for human head MRI. Body coil (volume resonator) has been widely equipped in 3T MRI systems for whole-body standing wave (SW) excitation. The wavelength λ_{head} (at 128MHz) in biological tissues is larger than the human head. The magnitude variation $\cos(k \cdot r)$ of SW is trivial in the human head (a). In comparison, local birdcage coil (volume resonator) has been industry standard for human head imaging at 7T. Since the wavelength λ_{head} (at 297MHz) in biological tissues approaches human head dimension, node and antinode of SW appear in the human head. The human body and the inner metallic surface of the MRI bore (the waveguide) become electrically large, so the volume resonator can radiate power inside, and the waveguide carries TE_{11} mode travelling wave (TW) at 297 MHz (b). However, the human body introduces discontinuity in wave impedance and phase velocity in the waveguide, leading to reflected power and secondary SW (c). To this end, dielectric waveguide has been proposed to achieve efficient TW excitation through TE_{11} -to- TM_{11} mode conversion, power focusing, wave impedance match, as well as phase velocity match (d).

References

- [1] Young, G. S., Kimbrell, V., Seethamraju, R. & Bublick, E. J. Clinical 7T MRI for epilepsy care: Value, patient selection, technical issues, and outlook. *Journal of Neuroimaging* 32, 377–388 (2022).
- [2] Shaffer, A. et al. Ultra-High-Field MRI in the Diagnosis and Management of Gliomas: A Systematic Review. *Front. Neurol.* 13, 857825 (2022).
- [3] Clarke, W. T. et al. Multi-site harmonization of 7 tesla MRI neuroimaging protocols. *NeuroImage* 206, 116335 (2020).

Comment #3

My other note, which I will frame as advice to the authors, is that they have not made it easy for reviewers like me to appreciate the work they have put in. Their written response to many of my original concerns is simply “we disagree,” followed by

reference to very general principles or rehashing of the Bloch equations. The authors are, of course, entitled to disagree with anything a reviewer says. On the other hand, a reviewer (especially one well versed in the Bloch equations not to mention the practical realities of high-field MR) is also entitled to the opinion that the authors have not taken their response seriously. I confess that the rebuttal document put me off enough that I almost missed the substantive changes to the manuscript (e.g., its new focus on efficiency as well as homogeneity, its extensive new field plots and explanations of mode conversion, etc.). Meanwhile, various features of the revised manuscript make it more difficult to review than it needs to be. The changed sections are all highlighted uniformly in yellow, making much of the new manuscript a sea of yellow. At least some marginal references to particular reviewer comments would have been helpful. Also, the figures, while informative, are quite busy, and more guidance in the captions about key features to note, or even about the meaning of various symbols, would also have been welcome. Figures are also cited out of order in several places. Grammar and syntax, especially in the revised sections of text, continue to require some editing.

Authors' Response:

We apologize deeply for our inappropriate ways in responding questions and in organizing revised contents, and meanwhile we feel very grateful to these valuable advices received. As instructed, revised content has been listed in the response to each comment; revised elaborations have been added in all figure captions with improved illustrations of key features, while mistakes in figure citations have been corrected; grammar and syntax have been improved throughout the manuscript.

Comment #4

Overall, I am more enthusiastic about the revised manuscript than I was about the original. I am still not convinced that traveling wave systems with subwavelength dielectric waveguides will rescue UHF MRI from challenges of B1+ homogeneity or efficiency – or even that UHF MRI needs as much rescuing as the authors claim it does.

Authors' Response:

We agree with that our preliminary research findings of dielectric waveguide TW method cannot rescue UHF MRI from challenges in B1+ homogeneity or efficiency, which, after all, has been an unmet need for decades.

The primary purpose of our work is to demonstrate the potential of a well-controlled TW system in addressing above challenges. Principles in designing travelling-wave transmission system (loaded with the human body) have been discussed in the present study including, wave-impedance match, TE-to-TM mode conversion, power-focusing, and phase-velocity match, while to the best of our knowledge, the later three of which have been firstly introduced in TW MRI.

At the same time, importantly, In addition, the principles of proposed guided travelling wave operation are well compatible with the state-of-art multi-channel transmission system [1-3], and the combination of both techniques hold certain promises in human whole-body imaging, which by all means deserves continuous exploration in future.

Following the reviewer's comment, overclaims in original text as shown below, have been removed in the revised manuscript.

[Abstract]

Our investigation offers insights into the design of new-generation TW MRI at UHF.

[Introduction]

Therefore, a TW transmitter operating under single-channel-transmission mode hold promises to be a feasible solution at UHF.

[Section: Subwavelength dielectric waveguide design for human head TW MRI at 7T]

The electrical length of the human body at operation frequency 297 MHz determines that the classic quasi-static excitation method is difficult to produce uniform B1+ over the entire human body.

[Summary]

The modified TW MRI method may advance one step further towards a feasibly and effective solution for whole-body MRI at UHF under single-channel-transmission mode, which, however, has remained as an unresolved technical challenge for over three decades.

References

- [1] Hoffmann, J., Mirkes, C., Shajan, G., Scheffler, K. & Pohmann, R. Combination of a multimode antenna and TIAMO for traveling-wave imaging at 9.4 Tesla. *Magn. Reson. Med.* 75, 452–462 (2016).
- [2] Elabyad, I. A. RF Shimming and Improved SAR Safety for MRI at 7 T With Combined Eight-Element Stepped Impedance Resonators and Traveling-Wave Antenna. *IEEE Trans. Microwave Theory Techn.* 66, 16 (2018).
- [3] Bluem, P., Van de Moortele, P.-F., Adriany, G. & Popovic, Z. Excitation and RF Field Control of a Human-Size 10.5-T MRI System. *IEEE Trans. Microwave Theory Techn.* 67, 1184–1196 (2019).

Comment #5

It is still not clear to me that modified traveling wave systems are more practical than simple birdcage transmit coils or, say, multicoil transmit arrays with fixed shims, which also require only a single transmit channel. However, I do find the principles of operation of the guided traveling wave system presented here to be thought-provoking,

and I expect I will not be alone in this.

Authors' Response:

We are very grateful to the reviewer's supportive comment.

Regarding the reviewer's concern, below is our thought and opinion:

At the operating RF frequency ($\geq 297\text{MHz}$) at UHF, equivalent wavelength in the human body is smaller than 20 cm, and the body size (assuming $170 \times 30 \times 20 \text{ cm}^3$) is becoming electrically large. As a result, well-delivered travelling wave is naturally more uniform than standing wave. According to aforementioned equations (1) and (2), the magnitude of travelling wave is ideally uniform in space, while the magnitude of standing wave " $\cos(\mathbf{k} \cdot \mathbf{r})$ " is a function of spatial coordinate \mathbf{r} as well as the wave number ($k = 2\pi/\lambda$).

Standing wave can appear in a loaded TW MRI system, and we have shown that how B1+ homogeneity improves through phase-velocity match, as indicated in revised Fig.6 (original Fig. 5) and Fig. 8 (original Fig. 7).

With the human body loaded, for the birdcage coil or multicoil transmit array designed in a volume shape (cylindrical current carrying surface), multiple incident waves entering the volume produce an intricate environment with complex wave interactions. Although EM fields can be manipulated through fixed shims, it is still difficult to achieve a large ($> \lambda/2$) uniform area at UHF [1-2]. While only small areas can be shimmed, shim settings are usually required to be updated on a subject-specific basis, which makes fixed shim less practical to use.

Besides, the TW approach also has benefit in its intrinsically lower SAR compared to birdcage coils or other resonators placed in the vicinity of the human body, since the human body is placed away from the reactive near-regions of the feed antenna for TW transmission. Recent studies have shown that reactive-field component is highly correlated with high SAR [3].

References:

- [1] Erturk, M. A., Li, X., de Moortele, P.-F. V., Ugurbil, K. & Metzger, G. J. Evolution of UHF Body Imaging in the Human Torso at 7T. *Topics in Magnetic Resonance Imaging* 28, (2019).
- [2] Sadeghi-Tarakameh, A. et al. In vivo human head MRI at 10.5T: A radiofrequency safety study and preliminary imaging results. *Magnetic Resonance in Med* 84, 484–496 (2020).
- [3] Solomakha, G. et al. A self-matched leaky-wave antenna for ultrahigh-field magnetic resonance imaging with low specific absorption rate. *Nature Communications* 12, 455 (2021).

REVIEWER #2

Comment #1

This study demonstrates an approach that can harness traveling wave with efficient implementation, which therefore provides important insights for the field development. The authors improved significantly the article with better description and characterization of the method. However, there are still two main concerns that needs addressing: 1) over the text the authors mention several times that the TW with single-channel transmission can be used as a solution for uniform B1 distribution at ultra-high field MRI. I don't think that this was demonstrated in previous works (as stated) and it was not demonstrated in this work. The authors achieved in this work similar RF efficiency and better homogeneity by 20% compared to a birdcage coil. Many other works also demonstrated that both efficiency and homogeneity can be improved with different RF coil designs. Although what authors showed here is an important achievement, it does not show that uniform distribution can be achieved with single channel transmission. This should be described more carefully in the text.

Authors' Response:

1. "... the TW with single-channel transmission can be used as a solution for uniform B1 distribution at ultra-high field MRI. I don't think that this was demonstrated in previous works (as stated)..."

(1)

Foo, et al. has shown that uniform B1+ in the radial plane can be achieved through manipulating the axial propagation constant k_z in a modified travelling-wave body coil driven by single-channel transmission [1]. In their study, the canonical volume resonator (body coil) was modified into the travelling wave mode through adding terminator resistors to eliminate power reflection, as shown in FIG. 2 in [1] (see figure below). Power splitter was connected to each leg to produce sinusoid current for uniform B1+ field.

FIG. 2. Diagram of the traveling wave test coil showing details of coil construction and placement of the terminating and matching resistors. The position of the saline phantom within the coil is also indicated by the dotted lines.

In such design, the EM field propagates as travelling wave along z-axis direction. Meanwhile, EM field in the radial plane still appears as standing wave due to its volume resonator design.

To alleviate B1+ inhomogeneity caused by the standing wave effect, the fundamental idea in Foo's study was to reduce radial propagation constant k_ρ , so as to obtain larger equivalent wavelength ($\lambda_p = 2\pi/k_\rho$) in the radial direction.

According to the relation $k_\rho^2 = k^2 - k_z^2$, radial propagation constant k_ρ can be lowered through increasing axial propagation constant k_z . Dielectric fillings in the coil-shield space were used to achieve a larger k_z . The propagation constant in the body (wave vector) $k = \omega^2 \mu \epsilon_0 \epsilon_r - j\omega \mu \sigma$ is determined by the permittivity and conductivity of the body. With lowered equivalent wavelength in radial directions, less variation of B1+ magnitude in the radial direction can be observed, as shown in FIG. 4 in [1] (see figure below).

FIG. 4. Radial field amplitude profiles for different dielectric material in the coil-to-shield space in a full-sized body coil at 64 MHz. The relative permittivity of the material is denoted by ϵ_{sh} .

(2)

The effectiveness of travelling-wave excitation in producing uniform B1+ distribution has also been revealed and discussed by Brunner, et al. in the very first paper of TW MRI (TW propagating in a waveguide) [1], which was stated as:

“By causing the underlying field pattern to propagate through space, such phase variation reduces the variation of the field magnitude. Notably, the limiting case of a plane wave has a perfectly uniform magnitude at any wavelength.”

Uniform excitation was evaluated through experiments over both phantoms and in vivo human subjects, as shown in FIG. 3 and FIG. 4 in [2] (see figures below).

Figure 3 | Example of wave impedance matching in travelling-wave MRI.
a, Non-uniform coverage of two phantom bottles is caused by residual standing radio-frequency waves, as also shown by the dashed image intensity profile on the right. **b**, The standing waves can be suppressed using wave impedance matching and dissipation in a termination load (solid profile).

Figure 4 | In vivo results. **a**, Travelling-wave MRI of a human lower leg *in vivo*. **b**, Identical scan performed with a traditional resonant probe.

Above results suggested that additional wave manipulation methods are of vital importance in exploring the potential of TW in producing uniform B1+.

2. "... the TW with single-channel transmission can be used as a solution for uniform B1 distribution at ultra-high field MRI...and it was not demonstrated in this work...Although what authors showed here is an important achievement, it does not show that uniform distribution can be achieved with single channel transmission. This should be described more carefully in the text."

In this study, all proposed methods for efficient TW MRI are based on single-channel transmission. Two ports at the patch feeding antenna, as shown in revised Fig. 7(e, f), were interfaced with a 90° quad-hybrid to provide circular polarization. The dielectric waveguide insert served as a passive component.

In addition, two waveguide ports shown in revised Fig. 3 were set for mode conversion efficiency measurement. Three ports shown in revised Fig. 6e were set for transmission coefficient measurement.

Revised Fig. 9 presented uniform B1+ distribution achieved by single-channel transmission TW MRI with dielectric waveguide insert utilized. The prominent center-brightening phenomenon (Fig. 9f) caused by the standing-wave effect (when

using canonical birdcage resonator) was inhibited to a certain extent.

The B1+ distribution quantitatively measured on the head phantom (shown in revised Fig. 9e, g) was not very uniform, and it was because the relative permittivity (78) of the phantom material is higher than that of the human brain (40~60), which led to sub-optimal wave impedance match. Nonetheless, the B1+ distribution along x-direction was more uniform in TW transmission.

In the revised manuscript, relevant texts have been improved for a more clear and accurate expression as follows.

[Caption of revised Fig. 7e]

The modified single-channel transmission TW waveguide system with dielectric cubic array insert. It was fed with a classic two-port patch antenna, interfaced with a 90° quad-hybrid to provide circular polarization.

[Caption of revised Fig. 9]

Human brain MRI results by using a birdcage coil vs. modified TW system at 7T. **Both approaches were quad-driven under single-channel transmission mode ...**

3. “The authors achieved in this work similar RF efficiency and better homogeneity by 20% compared to a birdcage coil. Many other works also demonstrated that both efficiency and homogeneity can be improved with different RF coil designs.”

While other works based on phased array coils have shown improvement in B1+ efficiency and homogeneity, these array coils require multi-channel transmission system which has limited access to clinical scenarios due to complex SAR management. Meanwhile, to the best of our knowledge, few studies have shown that such effect can be effectively inhibited through optimizing birdcage resonators or other single-channel driven RF coils.

We agree with that our preliminary research findings of dielectric waveguide TW method cannot rescue UHF MRI from challenges in B1+ homogeneity or efficiency, which, after all, has been an unmet need for decades.

The primary purpose of our work is to demonstrate the potential of a well-controlled TW system in addressing above challenges. Principles in designing travelling-wave transmission system (loaded with the human body) have been discussed in the present study including, wave-impedance match, TE-to-TM mode conversion, power-focusing, and phase-velocity match, while to the best of our knowledge, the later three of which have been firstly introduced in TW MRI.

At the same time, importantly, In addition, the principles of proposed guided

travelling wave operation are well compatible with the state-of-art multi-channel transmission system [3-5], and the combination of both techniques hold certain promises in human whole-body imaging, which by all means deserves continuous exploration in future.

References:

- [1] 1. Foo, T. K., Hayes, C. E. & Kang, Y.-W. Reduction of RF penetration effects in high field imaging. *Magnetic Resonance in Medicine* 23, 287–301 (1992).
- [2] Brunner, D. O., De Zanche, N., Fröhlich, J., Paska, J. & Pruessmann, K. P. Travelling-wave nuclear magnetic resonance. *Nature* 457, 994–998 (2009).
- [3] Hoffmann, J., Mirkes, C., Shajan, G., Scheffler, K. & Pohmann, R. Combination of a multimode antenna and TIAMO for traveling-wave imaging at 9.4 Tesla. *Magn. Reson. Med.* 75, 452–462 (2016).
- [4] Elabyad, I. A. RF Shimming and Improved SAR Safety for MRI at 7 T With Combined Eight-Element Stepped Impedance Resonators and Traveling-Wave Antenna. *IEEE Trans. Microwave Theory Techn.* 66, 16 (2018).
- [5] Bluem, P., Van de Moortele, P.-F., Adriany, G. & Popovic, Z. Excitation and RF Field Control of a Human-Size 10.5-T MRI System. *IEEE Trans. Microwave Theory Techn.* 67, 1184–1196 (2019).

Comment #2

2) Another concern that there are still many details/descriptions in Figures and in the text that require correction.

Below the detailed/specific comments:

Abstract

The abstract is somewhat long, repeating similar statements of “uniform spatial disturbing that is a key..., then repeating it “Uniform excitation under single-channel-transmission mode has been an unmet need “, while the results or conclusions does not offer a solution. There is an improvement of 20% of B1 inhomogeneity. So, I suggest to remove some parts of the repeated statements.

Authors' Response:

Thanks for the suggestion. As instructed, repeated statements have been removed and the abstract has been revised to be concise.

[Abstract]

Magnetic resonance¹⁻³ is featured with its versatility in detecting fine structure and complex dynamics of matter and biological tissues, and thus it has been widely used across interdisciplinary applications in physics⁴, biology^{5,6} and medicine^{7,8}. To excite nuclei spins efficiently, it is crucial to use a circular-polarized transverse magnetic field (\mathbf{B}_1^+) that operates at the Larmor frequency⁹. Uniform spatial distribution of \mathbf{B}_1^+ inside the subject is key to producing favorable tissue contrast in magnetic resonance imaging

(MRI)¹⁰. Radio-frequency (RF) coils designed under quasi-static conditions^{11,12} may not produce satisfying uniform B_1^+ at ultra-high-field (UHF) MRI, and traditional resonator coils usually cause issues such as severe inhomogeneous B_1^+ in electrically large human bodies¹³, which has remained as a major challenge over decades. Although radiative excitation (e.g., travelling-wave, TW) has shown promise at UHF, it has apparent disadvantages such as with inadequate excitation efficiency^{14,15}. In this study, we present a novel approach with subwavelength dielectric waveguide insert to enhance TW MRI efficiency as well as B_1^+ homogeneity at 7T. Principles of manipulating wave behaviors including: 1) TE₁₁-to-TM₁₁ mode conversion, 2) power focusing, 3) wave impedance match, and 4) phase velocity match, have been investigated through numerical simulation and imaging experiment. The TW MRI efficiency over the entire human brain was improved by 114% compared to classic TW method; compared to an industry-standard resonator coil, $|B_1^+|$ inhomogeneity in the human brain was reduced by 21.9%, and prominent center-brightening phenomenon caused by the standing-wave effect in birdcage resonator was inhibited to a certain extent. The fundamental improvement in TW MRI with effective wave behavior manipulation may promote the electromagnetic architecture design of UHF MRI systems.

Comment #3

“For the wave impedance match capability of the dielectric waveguide guide, opposed to unmatched 20% power transfer efficiency, up to 70% stimulated power was delivered to the load under well-matched condition” – this is not very clear sentence.

Authors' Response:

The original sentence regarding wave impedance match performance has been deleted in the abstract. Only the result of TW MRI efficiency improvement was provided.

[Abstract]

The TW MRI efficiency over the entire human brain was improved by 114% compared to classic TW method.

Comment #4

Introduction

p.3 1.103 – “the advantage of TW excitation in producing uniform B1 has been discovered ...” – here the sentence is based on an article by Foo, however the article discusses RF field homogeneity in frequencies of 64 and 170 MHz. I am not sure at all that the homogeneity that was described is relevant for 297MHz. However, there are more recent works that showed that TW can be used to manipulate the distribution (targeting uniform distribution) by combined several TW modes (which is similar to multi-channel transmission with the SW approach).

Authors' Response:

The diameter of the object investigated in Foo's study was 40cm, while the human head dimension (radial) investigated in this study was ~15cm. Although the working RF frequency (64MHz and 170MHz) Foo's study was different and smaller than 297 MHz, the B1+ inhomogeneity effect also appeared as center-brightening phenomenon, which was due to comparable wavelengths vs. object dimensions.

Comment #5

p.3 l.110-118 – It would be useful to add here in the introduction two references that used similar to the current work approach (although without detailed analysis) - in one passive coils were used to improve local RF field distribution when combined with TW transmission (1) and in another a dielectric was added to increase the RF field in a target region (2).

1. X. Yan, X. Zhang, J C. Gore, and W. A. Grissom . Improved traveling wave efficiency in 7T human MRI using wireless local loop and dipole arrays. Proc. Intl. Soc. Mag. Reson. Med. 25 (2017), 4291.
2. R. Schmidt, A. Webb. Improving travelling wave efficiency at 7 T using dielectric material placed "beyond" the region of interest. Proc. Intl. Soc. Mag. Reson. Med. 24 (2016) . 3532

Authors' Response:

Thanks for the suggestion. These two references have been added in the revised manuscript.

Comment #6

p.4.126 "In MRI, nuclei spins are excited exclusively by circular polarized transverse magnetic field" – I still think that the sentence is misleading. So I suggest rephrasing it to something like "In MRI, nuclei spins are excited exclusively by the effective circular polarized transverse magnetic field" or In MRI, nuclei spins are excited exclusively by circular polarized component of the transverse magnetic field".

Authors' Response:

Thanks for the suggestion. The sentence has been rephrased as "In MRI, nuclei spins are excited exclusively by circular polarized component of the transverse magnetic field".

Comment #7

Fig.2 The images in d show some permittivity values, while the plot in b shows different

set of values. It is somewhat confusing. Was there some reason to show different values? The plots in c show dependence in wall thickness. It would be useful to mention that it is shown in Fig.1d (as far as I understand). And what was the permittivity in c plot. What was D in plot b and for images in d.

Authors' Response:

We apologize for such confusing information provided in the figures. Accordingly, E field plots corresponding to relative permittivity of 16, 22, 26 have been presented in revised Fig. 3d (original Fig. 2). The relative permittivity in plot c and the wall thickness (D) used in plot b and d have been indicated in the caption of revised Fig. 3.

Fig. 3 The efficiency of TE₁₁-to-TM₁₁ mode conversion. Measured by transmission efficiency S_{12} of a two-port waveguide system. The empty part of the metallic waveguide acts as a filter in which only TE₁₁ mode can propagate (a). The dependency of mode conversion efficiency with relative permittivity ϵ_r (b) as well as wall thickness D of the dielectric waveguide (c). The electric field distributions in the metallic waveguide are reshaped by dielectric waveguide insert (d). The TE₁₁-to-TM₁₁ mode conversion efficiency is indicated from the residual electric field (TE₁₁) in the remaining part of the metallic waveguide without dielectric fillings, where TM₁₁ mode cannot propagate. The wall thicknesses in (b, d) are kept the same as 28 mm, while the relative permittivity ϵ_r in (c) is kept the same as 21.

Comment #8

Fig.3 no distances are shown in a and b), so it is hard to appreciate where are the relevant cross sections that are shown in c.

Authors' Response:

We apologize for missing important information. Scale bar has been provided in revised Fig. 4 (original Fig. 3) as shown below.

Figure 4

Comment #9

p.6 l.169 – “distilled water tubes with high dielectric constant “ – I suggest to make it more clear that this was done in another study, not in the current study, otherwise it is not clear here.

Authors’ Response:

We apologize for unclear description. Correct information has been provided in the revised manuscript as follows.

[Section: Power focusing]

In a previous study, distilled water tubes with high dielectric constant were used as waveguide fillings to improve TW MRI efficiency.

Comment #10

p.8.l.199 “Here, we demonstrate the dielectric waveguide structure can also be used for wave impedance matching” – “we demonstrate that”

Authors’ Response:

Thanks for the suggestion. It has been corrected in the revised manuscript.

Comment #11

p.8.1.204 “is lowered through increasing the length”- needs English editing

Authors' Response:

The sentence has been revised as instructed.

[Section: Wave impedance match]

The wave impedance match condition varies with the length of dielectric waveguide. In addition to minimizing power reflection, the dielectric waveguide has demonstrated the ability in reducing radiation loss. As a result, the power delivered to the load showed to be improved from 15% to 70%.

Comment #12

p.9.1.215 – “incoherent wave front ...triggers ... and leads to the formation SW” – maybe “formation of SW” or “formation of the SW”. It requires English editing.

Authors' Response:

The sentence has been revised as instructed.

[Section: Phase velocity matching]

As a result, incoherent wavefront formed at the air-tissue boundary triggers opposing propagating waves and resulted in the formation of SW within electrically large human bodies.

Comment #13

Fig.6 – “multiple dielectric cubes” – did you use different numbers in different cases. If so, please, write how many cubes were used for each case or write here instead of multiple – how many cubes were used here.

Authors' Response:

Thanks for the suggestion. The information of number of cubes (6×2) has been provided in the caption of revised Fig. 7 (original Fig. 6).

Comment #14

In the image it says that the permittivity was 21 and 52 and in the text 21 and 55. Fig.5 shows results for permittivity of 25 and 55 . Is it also a mistake or why different figures

show different values. I would suggest that for consistency, if the optimal solution was based on permittivity of 21 to show that case in all figures.

Authors' Response:

Thanks for pointing it out. There was a mistake in the relative permittivity value shown in the caption of original Fig. 6 (revised as Fig. 7), which has been corrected in the revised manuscript.

Fig. 7 Subwavelength dielectric waveguide for human head TW MRI at 7T. 6×2 dielectric cubes were used to constitute the hollow dielectric waveguide to provide higher structural flexibility (a). The EPE foam was used to attach cubic array along a cylindrical surface (b, c). The electric field distribution in a dielectric cylinder ($\epsilon_r=21$) inserted into the circular metallic waveguide, and the electric field distribution in a dielectric cubic array ($\epsilon_r=52$) inserted into the circular metallic waveguide (d). **The modified single-channel transmission TW MRI waveguide system with dielectric cubic array insert. It was fed with a classic two-port patch antenna, interfaced with a 90° quad-hybrid to provide circular polarization.** (e). The schematic and the photograph of the quad-driven feeding antenna (f). The transverse magnetic field component \mathbf{H}_{xy} vs. the longitudinal magnetic field component \mathbf{H}_z in the empty circular metallic waveguide (g, left) and in a dielectric waveguide inserted into the circular metallic waveguide (g, right). The transmission coefficient between the magnetic probe placed inside dielectric waveguide and the feeding antenna (h), calculated and measured in numerical simulations (S31, S32) and in MRI experiments (S31', S32'), respectively.

The relative permittivity values shown in revised Fig. 6 (original Fig. 5) and Fig. 7 (original Fig. 6) are supposed to be different. Because the dielectric waveguide shown in Fig. 6 is a hollow cylinder, while the dielectric waveguide in Fig. 7 was designed as cubic array for easy manufacturing and fine adjustment. Also, the load used in Fig. 6 was a lossless dielectric rod for better demonstration of the phase velocity match. In Fig. 7, the cubic array was designed for human head imaging.

Comment #15

In the figure caption it says S31, S32 and in the image it says S13, and so on.

Authors' Response:

Thanks for pointing it out. Such mistake has been corrected in revised Fig. 7 (original Fig. 6).

Comment #16

p.10.1.238 “As discussed above, TW excitation is a promising solution to produce uniform B1”. It was not really discussed above. As far as I understand to achieve uniform (or close to it) distribution one needs to combine several modes as was demonstrated in previous works with TW, however this is somewhat equivalent and similar in its complexity to the multi-channel transmit methodology that is currently developed.

Authors' Response:

Thanks for the question. Discussion about the natural strength in uniformity of TW excitation has been added in the introduction section. A new figure (Fig. 1) was drawn and has been added in the revised manuscript to help illustrate the underlying mechanism of travelling wave in producing uniform B1+.

[Introduction]

Compared to the SW, the travelling wave (TW) is naturally uniform in magnitude in space (shown in Fig. 1). For a normalized traveling wave f_{TW} with angular frequency ω , it can be explicitly expressed as the function of spatial coordinate \mathbf{r} and time t : $f_{TW}(\mathbf{r}, t) = \sin(\omega \cdot t + \mathbf{k} \cdot \mathbf{r})$, while a standing wave f_{SW} can be expressed as the superposition of two TWs propagating in opposite directions: $f_{SW}(\mathbf{r}, t) = f_{TW}^+ + f_{TW}^- = \sin(\omega \cdot t) \cdot \cos(\mathbf{k} \cdot \mathbf{r})$. It is clearly to see that the magnitude of TW is a constant. In contrast, the magnitude of SW varies in space as the function of $\cos(\mathbf{k} \cdot \mathbf{r})$, where $k=2\pi/\lambda$ is the wave number along propagation direction, λ is the wavelength. B1+ inhomogeneity residing in SW scales up with increased working frequency.

Fig.1. Schematic diagrams of RF transmission systems at 3T and 7T for human head MRI. Body coil (volume resonator) has been widely equipped in 3T MRI systems for whole-body standing wave (SW) excitation. The wavelength λ_{head} (at 128MHz) in biological tissues is larger than the human head. The magnitude variation $\cos(k \cdot r)$ of SW is trivial in the human head (a). In comparison, local birdcage coil (volume resonator) has been industry standard for human head imaging at 7T. Since the wavelength λ_{head} (at 297MHz) in biological tissues approaches human head dimension, node and antinode of SW appear in the human head. The human body and the inner metallic surface of the MRI bore (the waveguide) become electrically large, so the volume resonator can radiate power inside, and the waveguide carries TE_{11} mode travelling wave (TW) at 297 MHz (b). However, the human body introduces discontinuity in wave impedance and phase velocity in the waveguide, leading to reflected power and secondary SW (c). To this end, dielectric waveguide has been proposed to achieve efficient TW excitation through TE_{11} -to- TM_{11} mode conversion, power focusing, wave impedance match, as well as phase velocity match (d).

Comment #17

p.11 l.250 “multiple dielectric cubes” – please, write here how many cubes were placed.

Authors’ Response:

The information of number of cubes (6×2) has been provided in the revised manuscript.

Comment #18

p.12 l.283 – 287 “It should be noted ...” – very long sentence that is hard to follow. I would suggest to rephrase.

Authors’ Response:

The sentence has been rephrased as follows.

[Section: Subwavelength dielectric waveguide design for human head TW MRI at 7T]
It should be noted that compared to reactive coupling methods, radiative transmission methods have intrinsic lower filling factor in practice. Consequently, a greater proportion of power is dissipated in feed antenna when employing TW excitation as opposed to the resonator excitation. Conversely, higher filling factors result in increased SAR due to the amplified power dissipation in load. Hence, the use of SAR normalized efficiency is preferred for accurate quantitative assessment of the transmission efficiency.

Comment #19

Methods

p.14 1.348-350 – First sentence here describes the dielectric waveguide array with permittivity of 52 and the next sentence say that “all water cubes”. Water has permittivity of 78. I assume it was not water cubes, but a suspension of water and sucrose as was mentioned in other parts of the texts. So “water cubes” is misleading.

Authors' Response:

Thanks for pointing it out. It has been corrected in the revised manuscript.

[Section: Method-Numerical simulations]

As shown in Fig. 7a-c, all dielectric cubes were arranged on a cylindrical surface...

Comment #20

In “Subwavelength dielectric waveguide prototype” section:

First paragraph says that the cubes were filled with water. But it also says that sucrose was used to lower its permittivity. But it does not say to lower to what and what was the different materials percents in the mixture, as well as what were its measured final permittivity and conductivity. Then the next paragraph is highly confusing, since it repeats about the containers – are these different containers? But then it does not mention sucrose anymore.

Authors' Response:

We apologize for such confusing descriptions. These sentences have been rephrased thoroughly, and the information of water-sucrose mass ratio and measured permittivity and conductivity have been provided in the revised manuscript.

[Section: Method- Subwavelength dielectric waveguide prototype]

Each cube was manufactured as a polycarbonate container filled with sucrose-water solution. Distilled water was used to achieve low conductivity. Its frequency-selective response can be measured by the power transmission coefficient between a waveguide feed antenna and a magnetic field probe located inside the dielectric cylinder (as shown in Fig. 7e). The water/sucrose mass ratio of 100:54 was used, in order to achieve optimal power transmission as indicated in S31' at 297MHz (measured by a portable VNA), and the relative permittivity and conductivity were measured as 70 and 0.02 S/m respectively.

In total, 12 cubic polycarbonate containers...were filled with sucrose-water solution...

REVIEWER #3

Comment #1

I would like to thank the authors for their thorough review. However, although their results showed a slight increase in efficiency and homogeneity over previous travelling wave approach with CP excitation, the increase in performance is marginal in my belief.

Authors' Response:

We agree that compared to previous wave impedance matched travelling-wave (TW) approach, the proposed method showed slight increase in the performance of B1+ efficiency as well as SAR-normalized B1 efficiency.

However, previous TW approach was incapable of alleviating standing-wave through phase-velocity matching. Therefore, its B1+ inhomogeneity, measured as normalized RMSE of B1+ magnitude, was 30% higher than that of the proposed TW method with dielectric waveguide insert (as shown in Supplement Fig. 2). In previous approach, there was apparent low B1+ magnitude regions with significant magnitude variation at the back of the head, which is adverse to its practical applications.

As presented in this study, our approach can simultaneously enhance TW MRI excitation efficiency and homogeneity, which is an important improvement compared to previous TW approach.

Comment #2

The authors take a CP birdcage or CP mode as benchmark but this is not the state-of-the-art. Over the last decades multi-transmit 7T head coil technology has evolved and nowadays an 8 channel head coil with RF shimming is a standard procedure and provides superior B1+ control than shown here. Many of the workflow related issues have been largely solved for multi-channel 7T head imaging. It does not warrant a large investment into travelling mode direction anymore...

Authors' Response:

Indeed, we agree that the extensive innovative work devoted in multi-channel transmission has been important and crucial for RF engineering at ultra-high fields, which has been beneficial and inspiring to us as well. We proposed to resolve B1+ issue through the single-channel-transmission solution, but without any intention to diminish the great efforts that have been paid in multi-channel transmission development.

The CP birdcage [1-2] or CP mode [3] has been usually used as the benchmark for evaluation of novel phased array coils due to its superior performance at quasi-static regions. Meanwhile, the CP birdcage coil is the only standard coil allowed for clinical

7T human head imaging, and thereby it was used as benchmark in this study.

With superior B1+ control capability, however, SAR concern exists in multi-channel transmission, since the SAR distribution varies on a subject-specific basis. Therefore, elevated SAR risk as well as indirect SAR management schemes make multi-channel transmission not allowed in clinical scenarios for the time being.

Our study has shown that the B1+ inhomogeneity challenges (well-known as center-brightening phenomenon in the human head) can be feasibly inhibited through single-channel TW transmission. In addition, the principles of proposed guided travelling wave operation are nonetheless well compatible with the state-of-art multi-channel transmission system [4-6], and the combination of both techniques hold certain promises in human whole-body imaging, which by all means deserves continuous exploration in future.

References

- [1] Pfaffenrot, V. et al. An 8/15-channel Tx/Rx head neck RF coil combination with region-specific B₁ + shimming for whole-brain MRI focused on the cerebellum at 7T. *Magnetic Resonance in Med* 80, 1252–1265 (2018).
- [2] Clément, J., Gruetter, R. & Ipek, Ö. A combined 32-channel receive-loops/8-channel transmit-dipoles coil array for whole-brain MR imaging at 7T. *Magnetic Resonance in Med* 82, 1229–1241 (2019).
- [3] Feinberg, D. A. et al. Next-generation MRI scanner designed for ultra-high-resolution human brain imaging at 7 Tesla. *Nat Methods* 20, 2048–2057 (2023).
- [4] Hoffmann, J., Mirkes, C., Shajan, G., Scheffler, K. & Pohmann, R. Combination of a multimode antenna and TIAMO for traveling-wave imaging at 9.4 Tesla. *Magn. Reson. Med.* 75, 452–462 (2016).
- [5] Elabyad, I. A. RF Shimming and Improved SAR Safety for MRI at 7 T With Combined Eight-Element Stepped Impedance Resonators and Traveling-Wave Antenna. *IEEE Trans. Microwave Theory Techn.* 66, 16 (2018).
- [6] Bluem, P., Van de Moortele, P.-F., Adriany, G. & Popovic, Z. Excitation and RF Field Control of a Human-Size 10.5-T MRI System. *IEEE Trans. Microwave Theory Techn.* 67, 1184–1196 (2019).

Reviewer #1 (Remarks to the Author):

The authors' revisions have addressed my principal concerns.

Reviewer #2 (Remarks to the Author):

The authors further improved the article with better description and characterization of the method. My main concern is that although the authors described similar transmit efficiency of the described TW excitation and SW, the measurement was performed with a reference amplitude of 400 V for TW and 100 V for the SW. It will be good to explain the reason for these differences, and discuss whether currently it limits to achieve refocusing pulses.

In addition, few detailed comments are added below. Also, I think that additional English editing is required.

Below the detailed/specific comments:

Although, the work that authors present is important and can add new capabilities and push further the ultra-high field MRI, there are several works that also showed that new RF coil designs can improve homogeneity without the need to use B1 shimming or parallel transmit capabilities of the multi-channel transmit coils. Here some examples that can be mentioned, that improve the B1 homogeneity in the range of 10-40% without actual use of multi-channel coils different phase/amplitudes, but only due to the different design of the coil:

Krishnamurthy, N., Santini, T., Wood, S., Kim, J., Zhao, T., Aizenstein, H. J., & Ibrahim, T. S. (2019). Computational and experimental evaluation of the Tic-Tac-Toe RF coil for 7 Tesla MRI. *PloS one*, 14(1), e0209663.

Hossain, S., Taracila, V., Robb, F. J., Moore, J., & Winkler, S. A. (2023, July). Hardware Requirements for 2D Cylindrical-High Pass Ladder Coil Design enabling Homogeneous Excitation in Ultra High-Field MRI. In *2023 IEEE Symposium on Industrial Electronics & Applications (ISIEA)* (pp. 1-4). IEEE.

Avdievich, N. I., Nikulin, A. V., Ruhm, L., Magill, A. W., Henning, A., & Scheffler, K. (2022). Double-row dipole/loop combined array for human whole brain imaging at 7 T. *NMR in Biomedicine*, 35(10), e4773.

Avdievich, N. I., Solomakha, G., Ruhm, L., Nikulin, A. V., Magill, A. W., & Scheffler, K. (2021). Folded-end dipole transceiver array for human whole-brain imaging at 7 T. *NMR in Biomedicine*, 34(8), e4541.

Line 46 "such as with inadequate excitation efficiency" – maybe "such as an inadequate excitation efficiency"

Line 63-64 $B_1 = (H_x + iH_y)/2$. Please, check this equation, permeability is missing here. The same in Figure 1.

Line 89-90 I suggest to rephrase the sentences that describe the TW with magnitude naturally uniform. I am not sure what it means, since the magnitude depends on the reflected fields and the tissues boundaries. As was shown in the article by Brunner D. at 2009, in large samples (as the body), there can be field perturbations, due to different tissues or boundaries, as shown in Fig,5b. Similarly, as shown by the authors in Figure 8, after phase velocity match, the resulting B1 distribution is still clearly not homogeneous. So, I suggest to rephrase.

Line93 "it is clearly to see that" – needs English editing.

Line 129 "Therefore, efficient TW excitation requires propagating transverse magnetic modes dominantly." – requires English editing.

Fig.5 – I suggest to move the x,z arrows to be shown from the left side of the images, otherwise, it looks more like it is related to the figure in c.

Line 192 "were efficiently excited, as composed to a large area ..." – "as composed" – you meant "as opposed to"?

Figure 8 – the figure caption does not explain what is the difference between the two cases shown in a and in b.

Line 275 “Resistive losses from both feed antenna and resonator were underestimated in numerical simulations” - the authors meant “estimated” instead of “underestimated”?

We thank the reviewers for their careful assessments of this study and their insightful and helpful comments. We have substantially revised the manuscript, and all revised sentences are printed in **Red** color in the manuscript. Below we address each of the review comments, and our responses are printed in **Blue**.

REVIEWER #2

Comment #1

The authors further improved the article with better description and characterization of the method. My main concern is that although the authors described similar transmit efficiency of the described TW excitation and SW, the measurement was performed with a reference amplitude of 400 V for TW and 100 V for the SW. It will be good to explain the reason for these differences, and discuss whether currently it limits to achieve refocusing pulses.

Authors' Response:

Thanks for the suggestion. Here we provide further elaborations on the relevant issues as follows.

1. Transmit efficiency measurement and calculation

We understand the concern about the transmit efficiency. In Fig.9 e&g, \mathbf{B}_1^+ maps were normalized to 1W power in load ($\mathbf{B}_{1W-load}^+$) instead of 1W input power ($\mathbf{B}_{1W-power}^+$), which can account for such disparity.

The classic \mathbf{B}_1^+ normalization method, with reference to 1W input power ($\mathbf{B}_{1W-power}^+$), was not considered for comparison in this study, because the birdcage resonator (SW) and radiative method (TW) work fundamentally differently in power delivering. The nature of radiative transmission in, i.e., lower filling factors and higher proportions of power loss from transmitters, leads to reductions in both of $|\mathbf{B}_1^+|$ and SAR. Therefore, SAR normalized transmit efficiency ($|\mathbf{B}_1^+|/\sqrt{\max. \text{SAR}}$) is preferred as a more reasonable quantitative assessment.

However, reliable quantitative SAR mapping method in MRI is still missing. Although MRI Thermometry can be used as an alternative method to indirectly evaluate SAR distribution in saline phantoms, conventionally equipped 8kW power amplifier paired with local resonators usually cannot guarantee adequate amount of power delivered to the load with TW transmission. Therefore, the temperate change may not be significant enough to be captured by MRI Thermometry.

Consequently, power loss in load was used to normalize B_1^+ in this study (B_1^+ normalized to 1W power in load). Such normalization can be achieved through measuring the quality factor ratio ($Q_{ratio}: \frac{Q_{unloaded}}{Q_{loaded}} = \frac{P_{Tx-loss}+P_{load-loss}}{P_{Tx-loss}}$), in which $P_{Tx-loss}$ is power loss from the transmitter (including waveguide in TW method), and $P_{load-loss}$ is power loss in load, while $Q_{unloaded}$ and Q_{loaded} can be measured using the standard double-probe method [1]. The patch antenna for TW excitation was placed at the service end of the MRI scanner bore for quality factor measurement. The normalization factor $Norm.1W-load$ was calculated as: $\frac{1}{\left(1-\frac{1}{Q_{ratio}}\right)} = \frac{P_{Tx-loss}+P_{load-loss}}{P_{load-loss}}$. Then, the B_1^+ normalized to 1W power in load can be obtained as: $B_{1\ 1W-load}^+ = B_{1\ 1W-power}^+ \cdot Norm.1W-load$. In this study, the Q_{ratio} of the patch antenna (waveguide side) and the birdcage resonator were measured as 1.03 and 2.5 respectively. The $Norm.1W-load$ for TW was 4.5 times larger than that of SW. Therefore, the TW showed similar transmit efficiency in $B_{1\ 1W-load}^+$. It agrees well with numerical simulation results in Fig. 9c, where the resistive power loss is usually underestimated.

According to the low Q_{ratio} of TW method measured in this study, the power loss from the transmitter (including waveguide) was much more dominant. It may attribute to the waveguide structure and material which consist of cryostat inner bore and gradient shield (metallic mesh).

2. The limit in achieving refocusing pulse

For efficient implementation of the TW in this study, 484V in transmit voltage would be required in order to ideally achieve 90° spin excitation in an anthropomorphic head phantom. Limited by the peak power (8kW) of the RF amplifier equipped on the 7T MRI scanner (MAGNETOM 7T, Siemens Healthcare, Erlangen, Germany) which was designed to be paired with local resonators, only up to 120° refocusing pulse can be achieved. Therefore, an upgraded power amplifier is preferred to achieve strict 180° refocusing pulses.

In the meantime, the low filling factor nature of TW method is comparable to body coil transmitters instead of local resonators, as shown in Fig. 1. For example, 1.5T MRI body coils are typically equipped with 15-20 kW power amplifiers, while 3T body coils are usually paired with 35kW power amplifiers, which can be attributed to elevated power loss during RF power transmission as well as dielectric loss in loadings [2]. It should be noted that body coils are used for whole-body excitation including the head and other small-size body parts.

Nonetheless, for the present study, 20-35kW power amplifiers would be adequate to achieve strict 180° refocusing pulses for the efficient implementation of TW at 7T.

Moreover, through substituting current embedded waveguide structures with low loss waveguides, the requirement for power amplifier can be further lowered.

According to the suggestion raised by the reviewer, detailed elaborations of transmit efficiency measurements and calculations, as well as discussions on current limits in achieving refocusing pulses have all been provided in the revised manuscript as follows.

[Section: Subwavelength dielectric waveguide design for human head TW MRI at 7T]
“... The classic B_1^+ normalization method, with reference to 1W input power ($B_{1\text{ }1W\text{-power}}^+$), was not considered for comparison in this study, because the birdcage resonator (SW) and radiative method (TW) work fundamentally differently in power delivering ...”

“... However, RF amplifier upgrade is preferred for a full implementation of TW transmission due to its intrinsically low $B_{1\text{ }1W\text{-power}}^+$. At present, conventionally equipped 8 kW power amplifiers which were designed to be paired with local resonators can only achieve up to 120° refocusing pulses.”

References:

- [1] Keil, B. & Wald, L. L. Massively parallel MRI detector arrays. *J. Magn. Reson.* 229, 75–89 (2013).
- [2] Vaughan, J. T. et al. Efficient high-frequency body coil for high-field MRI. *Magnetic Resonance in Med* 52, 851–859 (2004).

Comment #2

In addition, few detailed comments are added below. Also, I think that additional English editing is required.

Authors' Response:

Thanks for the comments and suggestion. English editing has been conducted throughout the manuscript as instructed.

Below the detailed/specific comments:

Comment #3

Although, the work that authors present is important and can add new capabilities and push further the ultra-high field MRI, there are several works that also showed that new RF coil designs can improve homogeneity without the need to use B1 shimming or parallel transmit capabilities of the multi-channel transmit coils. Here some examples that can be mentioned, that improve the B1 homogeneity in the range of 10-40%

without actual use of multi-channel coils different phase/amplitudes, but only due to the different design of the coil:

Krishnamurthy, N., Santini, T., Wood, S., Kim, J., Zhao, T., Aizenstein, H. J., & Ibrahim, T. S. (2019). Computational and experimental evaluation of the Tic-Tac-Toe RF coil for 7 Tesla MRI. *PloS one*, 14(1), e0209663.

Hossain, S., Taracila, V., Robb, F. J., Moore, J., & Winkler, S. A. (2023, July). Hardware Requirements for 2D Cylindrical-High Pass Ladder Coil Design enabling Homogeneous Excitation in Ultra High-Field MRI. In *2023 IEEE Symposium on Industrial Electronics & Applications (ISIEA)* (pp. 1-4). IEEE.

Avdievich, N. I., Nikulin, A. V., Ruhm, L., Magill, A. W., Henning, A., & Scheffler, K. (2022). Double-row dipole/loop combined array for human whole brain imaging at 7 T. *NMR in Biomedicine*, 35(10), e4773.

Avdievich, N. I., Solomakha, G., Ruhm, L., Nikulin, A. V., Magill, A. W., & Scheffler, K. (2021). Folded-end dipole transceiver array for human whole-brain imaging at 7 T. *NMR in Biomedicine*, 34(8), e4541.

Authors' Response:

Thanks for the comments and suggestions. Additional introduction along with references to these examples have been added in the revised manuscript as follows.

[Introduction]

“Recent advances in coupled-mode methods^{32,33} as well as power splitters³⁴⁻³⁷ have shown the feasibility of driving phased array resonators under the single-channel mode with improved B_1^+ homogeneity.”

Comment #4

Line 46 “such as with inadequate excitation efficiency” – maybe “such as an inadequate excitation efficiency”

Authors' Response:

Thanks for the suggestion. This sentence has been corrected as “...such as an inadequate excitation efficiency.”

Comment #5

Line 63-64 $B_1 = (H_x + iH_y)/2$. Please, check this equation, permeability is missing here. The same in Figure 1.

Authors' Response:

Thanks for pointing it up. The formula of B_1^+ has been corrected as $B_1^+ = \mu \cdot (H_x + iH_y) / 2$ in the revised manuscript.

Comment #6

Line 89-90 I suggest to rephrase the sentences that describe the TW with magnitude naturally uniform. I am not sure what it means, since the magnitude depends on the reflected fields and the tissues boundaries. As was shown in the article by Brunner D. at 2009, in large samples (as the body), there can be field perturbations, due to different tissues or boundaries, as shown in Fig.5b. Similarly, as shown by the authors in Figure 8, after phase velocity match, the resulting B1 distribution is still clearly not homogeneous. So, I suggest to rephrase.

Authors' Response:

Thanks for the suggestion. The sentence has been rephased accordingly as follows.

[Introduction]

“Compared to SW, travelling wave (TW) is naturally more uniform in magnitude when propagating through an infinitely large homogenous medium. For the simple format of a traveling wave $f_{TW} \dots$ ”

Comment #7

Line93 “it is clearly to see that” – needs English editing.

Authors' Response:

This sentence has been corrected as “It is clear that the magnitude of TW is a constant, while the magnitude of SW varies in space as the function of ...”.

Comment #8

Line 129 “Therefore, efficient TW excitation requires propagating transverse magnetic modes dominantly.” – requires English editing.

Authors' Response:

This sentence has been modified as “Therefore, transverse magnetic (TM) modes are preferred for efficient TW excitation ...”

Comment #9

Fig.5 – I suggest to move the x,z arrows to be shown from the left side of the images, otherwise, it looks more like it is related to the figure in c.

Authors' Response:

Thanks for the suggestion. Fig. 5 has been modified accordingly (shown below), with the x,z arrows moved to the left hand side.

Revised Figure 5

Comment #10

Line 192 “were efficiently excited, as composed to a large area ...” – “as composed” – you meant “as opposed to”?

Authors' Response:

Thanks for pointing it up. This typo has been corrected in the revised manuscript as “are efficiently excited, as opposed to exciting a larger area ...”

Comment #11

Figure 8 – the figure caption does not explain what is the difference between the two

cases shown in a and in b.

Authors' Response:

There is no difference in presented data in (a) and (b). The color range in (b) was re-adjusted from [0, 0.8] to [0, 0.25] to better showcase the efficacy of power focusing within the human head region. Additional explanation has been added in the revised figure caption of Fig. 8 as follows.

[Caption of Fig.8]

“...The distribution of B_1^+ magnitude within the human subject in an MRI-embedded waveguide is displayed in (b), with the color range re-adjusted to [0, 0.25] to better showcase the efficacy of power focusing within the human head region.”

Comment #12

Line 275 “Resistive losses from both feed antenna and resonator were underestimated in numerical simulations” - the authors meant “estimated” instead of “underestimated”?

Authors' Response:

It is “underestimated”, since the resistive power loss from RF coils is usually not well modelled in full-wave EM simulations as elaborated in [1], and additional bench measurements of quality factor are therefore necessary for corrections. To avoid misunderstanding, such expression has been modified in the revised manuscript as follows.

[Section: Subwavelength dielectric waveguide design for human head TW MRI at 7T]
“According to the simulation result, modified TW method with dielectric waveguide has shown comparable...It should be noted that, the resistive power loss from transmitters is underestimated in full-wave numerical simulations.”

References:

[1] Gao, Y. & Zhang, X. Intrinsic Temporal Performance of the RF Receive Coil in Magnetic Resonance Imaging. IEEE Trans. Med. Imaging 41, 3432–3444 (2022).

Reviewer #2 (Remarks to the Author):

The authors' revisions have addressed my concerns.